# Atmospheric $CO_2$ effect on stable carbon isotope composition of terrestrial fossil archives

Vincent J. Hare [1,2], Emma Loftus[1,3], Amy Jeffrey[1] & Christopher Bronk Ramsey[1]

The $^{13}C/^{12}C$ ratio of $C_3$ plant matter is thought to be controlled by the isotopic composition of atmospheric $CO_2$ and stomatal response to environmental conditions, particularly mean annual precipitation (MAP). The effect of $CO_2$ concentration on $^{13}C/^{12}C$ ratios is currently debated, yet crucial to reconstructing ancient environments and quantifying the carbon cycle. Here we compare high-resolution ice core measurements of atmospheric $CO_2$ with fossil plant and faunal isotope records. We show the effect of $pCO_2$ during the last deglaciation is stronger for gymnosperms (−1.4 ± 1.2‰) than angiosperms/fauna (−0.5 ± 1.5‰), while the contributions from changing MAP are −0.3 ± 0.6‰ and −0.4 ± 0.4‰, respectively. Previous studies have assumed that plant $^{13}C/^{12}C$ ratios are mostly determined by MAP, an assumption which is sometimes incorrect in geological time. Atmospheric effects must be taken into account when interpreting terrestrial stable carbon isotopes, with important implications for past environments and climates, and understanding plant responses to climate change.

[1] Research Laboratory for Archaeology and the History of Art, School of Archaeology, University of Oxford, 1 South Parks Road, Oxford OX1 3TG, UK. [2] Department of Earth and Environmental Sciences, University of Rochester, Rochester, NY 14627, USA. [3] Merton College, University of Oxford, Merton Street, Oxford OX1 4JD, UK. Correspondence and requests for materials should be addressed to V.J.H. (email: vincent.john.hare@gmail.com)

At present the global mean stable carbon isotope composition of $C_3$ plants ($\delta^{13}C_p$), most of Earth's vegetation, is about −27‰ relative to VPDB, although $\delta^{13}C$ varies widely between about −22 and −36‰[1]. Understanding the sources of this variation has been the major aim of stable isotope studies of plant physiology and ecology[2–4]. Studies of modern plants[1, 5] have found that although correlations exist with variables that include plant functional type and altitude, $\delta^{13}C_p$ is most strongly correlated with mean annual precipitation (MAP). Values more positive than about −22‰ are found in arid and hyperarid regions, whereas values more negative than −31.5‰ are restricted to closed-canopy tropical forests. However, there is currently no accurate understanding of how plant $\delta^{13}C_p$ varies with past atmospheric $CO_2$ and climate. Nearly thirty years ago, a striking correspondence was first noted between a 4‰ change in the $\delta^{13}C_p$ of North American trees and the deglacial rise in atmospheric $CO_2$ concentration[6]. The difference, also identified in fossilised *Pinus flexilis* needles[7] and Japanese conifers[8], has been explained by changes in leaf water-use efficiency and stomatal conductance under conditions of changing $pCO_2$[7, 9, 10], which are derived from classical models of photosynthetic fractionation[11, 12]. The complexity of these models is seemingly at odds with experimental data from plant growth chambers obtained by Schubert and Jahren[13], who propose a simple alternative model which depends on changes in only two atmospheric variables, i.e. $pCO_2$ and the source isotopic composition of carbon dioxide ($\delta^{13}C_{CO_2}$). According to this simple model, most of the global change in $\delta^{13}C_p$ of fossil leaves and bulk terrestrial organic matter from the past 30 kyr[14] can be explained by the deglacial rise in $pCO_2$. This model has been disputed[15] on two bases: first, that the change in $\delta^{13}C_p$ can be completely explained by an increase in MAP, differential organic degradation, and changes in $\delta^{13}C_{CO_2}$, and second, that fossil collagen and tooth enamel from the Eocene to the historical era apparently do not discern a $pCO_2$ effect. On the other hand, globally averaged records of speleothem $\delta^{13}C$ appear to support a strong $pCO_2$ dependence over the past 90 kyr[16], and the issue is thus unresolved.

An accurate model of the factors which control $\delta^{13}C_p$ is of great importance for understanding $CO_2$ exchange during glacial–interglacial cycles[17], evaluation of palaeo-$CO_2$ proxies for timescales beyond the ice-core record[18], and investigating $CO_2$ assimilation by the biosphere under future anthropogenic emissions[19]. There are also fundamental implications for palaeoecology. The average $\delta^{13}C$ of modern $C_4$ plants is around −12.5‰[20], and the clear difference of ~14‰ between this and the values of $C_3$ plants gained early recognition as an effective method of distinguishing between the two photosynthetic pathways[21, 22]. This difference is also passed onto animal tissues, forming the basis for reconstructions of the diets of ancient fauna and hominins[20, 23, 24]. Predictions from models of photosynthetic fractionation, however, suggest that $C_3$ and $C_4$ plants respond differently to changes in global $pCO_2$, and the magnitude and timing of changes to $\delta^{13}C_p$ of each group will differ. Knowing the details of $\delta^{13}C_p$ response to changing atmospheric $CO_2$ is therefore crucial in the interpretation of faunal stable isotope records as proxies of $C_3$ and $C_4$ vegetation cover in mixed environments in deep time[25–27] as well as reconstructing environmental parameters such as MAP and/or forest cover in $C_3$ settings[28–30].

Here we address the issue by exploring the implications of changes in atmospheric $\delta^{13}C_{CO_2}$ and $pCO_2$ for $\delta^{13}C$ proxies from terrestrial archives of carbon. We use recently published high-resolution data for $\delta^{13}C_{CO_2}$ and $pCO_2$ from Antarctic ice cores to compute predictions of $\delta^{13}C_p$ under four different models of photosynthetic fractionation over the past 155 kyr. We then compare model predictions with two high temporal resolution

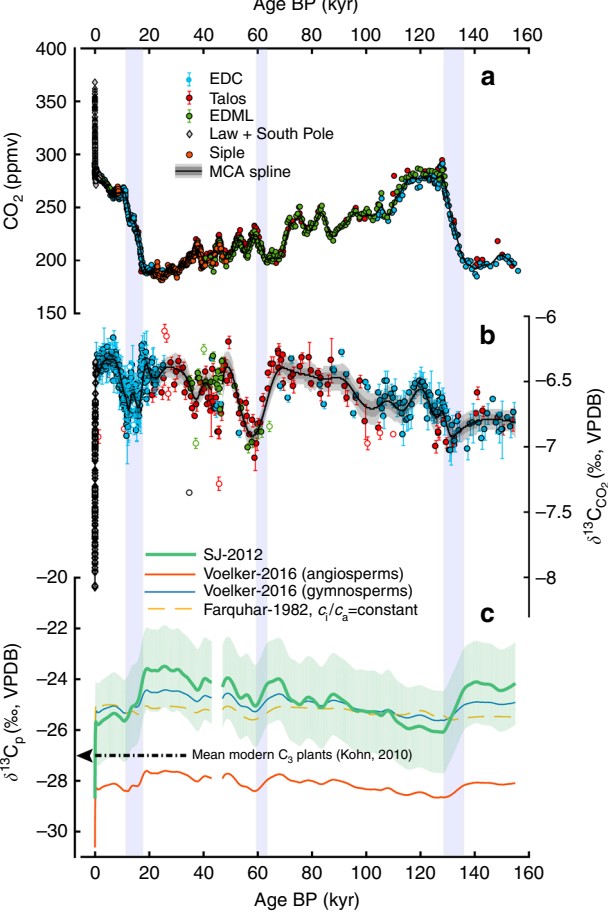

**Fig. 1** Ice core records of stable carbon isotope composition and concentration of $CO_2$ from 155 kyr BP to the present. **a** $CO_2$ concentration data from the preindustrial era compiled from EPICA Dome C (EDC), Talos Dome, EPICA Dronning Maud Land (EDML), and Siple ice cores, with post-industrial records from Law Dome and South Pole. Black curve shows a Monte Carlo average (MCA) spline. **b** Corresponding $\delta^{13}C_{CO_2}$ records and MCA spline. Error bars are 1σ uncertainties. Dark grey region is 1σ confidence interval for spline. Open circles indicate outliers identified by ref. [33]. **c** Models of $\delta^{13}C_p$ calculated from ice core records. SJ-2012 model (light green curve) is shown with 1σ propagated uncertainties. Blue bars indicate three periods where this model predicts a change in $\delta^{13}C_p$ of more than 1‰ at a rate exceeding 0.25‰/kyr, excluding the period of recent anthropogenic change

compilations of $\delta^{13}C$ from wood cellulose and faunal collagen which span the last deglaciation, to infer the relative magnitudes of changes due to MAP, $pCO_2$ and $\delta^{13}C_{CO_2}$.

## Results

**Photosynthetic fractionation over the last glacial cycle.** First, we mathematically model the effects of shifts in $pCO_2$ from 150 to 400 ppmv, over a range of relevant $\delta^{13}C_{CO_2}$ (glacial maxima to present day) to demonstrate the potential magnitude and direction of changes in $\delta^{13}C_p$. We consider four models which represent different scenarios of plant physiological control on isotope fractionation (Supplementary Fig. 1) in $C_3$ land plants. The first three models are based on the expression developed by Farquhar et al.[11, 12], which combines fractionations associated with carboxylation and diffusion of $CO_2$ into the leaf:

$$\delta^{13}C_p = \delta^{13}C_{CO_2} - a - (b-a)\frac{c_i}{c_a}, \qquad (1)$$

where $a$ is the magnitude of fractionation during gaseous diffusion of $CO_2$ through the lead boundary layer and stomata, and $b$ is the magnitude of net discrimination during carboxylation in $C_3$ plants. Both diffusion and carboxylation are dependent on the ratio of leaf intercellular to atmospheric partial pressures of $CO_2$ ($c_i/c_a$).

Two of these models assume that $c_i/c_a$ varies linearly with $c_a$, but with different gradients for angiosperms and gymnosperms, which are called models "Voelker-2016a" and "Voelker-2016g" respectively (see Methods for details). Motivation for different leaf gas-exchange strategies in these two groups comes from analyses of modern experiments and fossil tree-ring data from the Last Glacial[31], as well angiosperm leaf waxes and terpenoids from the Palaeogene[32], which consistently show a 2‰ depletion in $^{13}C$ relative to gymnosperm species. For comparison, another model (hereafter "Farquhar-1982") assumes a leaf gas-exchange strategy where constant $c_i/c_a$ is maintained across the entire range of $pCO_2$, which is unlikely. Finally, we consider the hyperbolic model (SJ-2012) proposed by Schubert and Jahren[13], which is based on the expression

$$\Delta^{13}C = [(A)(B)(pCO_2 + C)]/[A + (B)(pCO_2 + C)], \quad (2)$$

where $\Delta^{13}C = \left(\delta^{13}C_{CO_2} - \delta^{13}C_P\right)/\left(1 + \delta^{13}C_P/1000\right)$, and $A$, $B$, and $C$ are constants obtained by fitting Eq. (2) to data from modern growth chamber experiments conducted on *Raphanus sativus* and *Arabidopsis* plants, as well as fossil $\delta^{13}C_P$ from the Last Glacial, which together provides a large range of $pCO_2$ (180 to 4200 ppmv) and $\delta^{13}C_{CO_2}$ (−6.4 to −18.0‰).

To identify the timing and magnitude of expected shifts in $\delta^{13}C_P$ over the past 155 kyr, we compute the predictions of the four models using Antarctic ice core records of $\delta^{13}C_{CO_2}$ and $\delta^{13}C_{CO_2}$ (Supplementary Data 1). Our ice core compilation (Fig. 1) makes use of a recently published record[33] which significantly improves the temporal resolution of $\delta^{13}C_{CO_2}$ measurements between Terminations I and II, which are already well-represented. These $pCO_2$ and $\delta^{13}C_{CO_2}$ measurements present a near-continuous record, apart from the period between 47–43 kyr BP, where there is disagreement between EPICA Dronning Maud Land and Talos Dome cores.

All models, with the exception of Farquhar-1982, resolve a −2.5‰ change in $\delta^{13}C_P$ due to the anthropogenic isotope effect, and offer similar predictions for future $\delta^{13}C_P$. However, it is important to note that the models diverge strongly under conditions of low $pCO_2$. During the period between the Last Glacial and the beginning of the Holocene (11.4 kyr BP), ice cores document an 80 ppmv rise in $pCO_2$, which is accompanied by fluctuations in $\delta^{13}C_{CO_2}$ of up to 0.3‰ (Fig. 1). Over the same period, a change of similar magnitude (i.e. 0.3‰) is predicted in $\delta^{13}C_P$ by Farquhar-1982, which assumes a negligible $pCO_2$ effect. The Voelker-2016a and Voelker-2016g models predict a larger change in $\delta^{13}C_P$ of −0.8‰ and −0.9‰, respectively. The largest change is predicted by SJ-2012 (−2.0‰), which is comparable to the recent anthropogenic isotope effect. This is a significant change which is larger than that implied by an hypothetical doubling in MAP (1‰)[15], combined with any changes in $\delta^{13}C_{CO_2}$ (<0.3‰) over the LGM/Holocene transition.

Furthermore, the SJ-2012 model predicts high amplitude changes (>1‰) in $\delta^{13}C_P$ during three periods over the past 155 kyr (12–18, 60–62.7, and 129.4–135 kyr). These changes occur at a rate which exceeds 0.25‰/kyr (Fig. 1), excluding the past 130 years, where the change in $\delta^{13}C_P$ is two orders of magnitude greater (40‰/kyr). The durations of the three pre-Industrial high-amplitude episodes range from 2.7 to 5.6 kyr, and are therefore relatively brief on Quaternary timescales. Interestingly, while two of these episodes can be attributed to the rise in

$pCO_2$ during both glacial terminations, the 1‰ shift in $\delta^{13}C_P$ predicted by this model between 60 and 62.7 kyr is mainly driven by a 0.5‰ decrease in $\delta^{13}C_{CO_2}$, which is accompanied by an increase in $CO_2$ concentration occurring during Marine Isotope Stage (MIS) 4[33].

The rates and timings of these predicted changes need to be considered when evaluating the $pCO_2$ effect from fossil archives. Previous examination of faunal collagen and tooth enamel from the Eocene to the present appears to show no $pCO_2$ effect[15]. However, considering that our analysis shows that high amplitude changes in $\delta^{13}C_P$ are predicted to occur during relatively brief periods (i.e. ~2.7–5.6 kyr), and faunal data from previous studies are thinly represented across several million years, it is unlikely that they provide the necessary temporal resolution to discern a possible $pCO_2$ effect. In other words, beyond the limit of radiocarbon dating (~50 kyr), fossil archives will have minimum age uncertainties of several thousand years, which makes evaluation of the $pCO_2$ effect impossible. Our analysis reveals that the only period during which the effect would be statistically distinguishable is the last deglaciation, when radiocarbon methods offer sufficient dating precision (~50–300 yr, $1\sigma$).

**Comparison with plant and faunal isotope archives.** To test each model we compile a record of $\delta^{13}C_P$ which is based on 720 samples of radiocarbon-dated wood cellulose from the Northern Hemisphere, spanning the last deglaciation (Fig. 2). We also compile 521 $\delta^{13}C$ values of well-dated herbivore collagen from predominantly $C_3$ locations in northwestern Europe and northern Eurasia. Since this compilation is biased towards these regions, and herbivore diets are selective, the faunal record will not always reflect the 'average' composition of plants in an eco-system. Additionally, our cellulose data are over-represented by woody species from temperate northern latitudes. Therefore, to make these very widely dispersed samples comparable, we adopt a strategy of adjusting both cellulose and collagen $\delta^{13}C$ records for geographical variability in latitude, altitude and MAP (see Methods). When we adjust for geographical variability in this way (Fig. 2) the amplitude of changes across the LGM/Holocene transition is reduced from 1.41 to 0.93‰ (fauna) and 3.54 to 2.77‰ (plants). Therefore, the residual effect appears different for both cellulose and collagen. Note that our faunal data show greater scatter than our cellulose records, but the shift in $\delta^{13}C$ between <10 and >20 kyr faunal data is statistically significant (one-sided paired sample $t$-test, $t = −5.9$, $p = 5.2 \times 10^{−8}$, $\alpha = 0.05$).

We hypothesise that the residual isotopic changes in adjusted cellulose and collagen records through time consist of two components: first, changing atmospheric chemistry, and second, changes in MAP, which are reasonably described by the regressions of Kohn[1]. Under this assumption we are able to constrain the $pCO_2$ effect after correcting adjusted $\delta^{13}C_P$ for the effect of changing MAP between <10 and >20 kyr, which we infer from an ensemble of coupled atmosphere-ocean general circulation models (GCMs, see Methods for further details). Note that these corrections are different from our adjustments for geographical variability; whilst the latter only adjust for geographical bias, the former correct for changes through time. We find that in most cases (~90% for fauna, ~95% for plants) the PMIP3-CMIP5 ensemble model predicts a change to wetter conditions during the Holocene. The average effect of MAP on the isotopic signal, constrained by GCMs, is therefore negative for both plants and fauna (Fig. 3b, c). The effect of faunal records, confined to Eurasia, is $−0.4 \pm 0.88$‰, which is larger than both gymnosperms ($−0.27 \pm 0.55$‰) and all plants ($−0.13 \pm 0.74$‰) but smaller than plants from North America ($−0.46 \pm 0.88$‰).

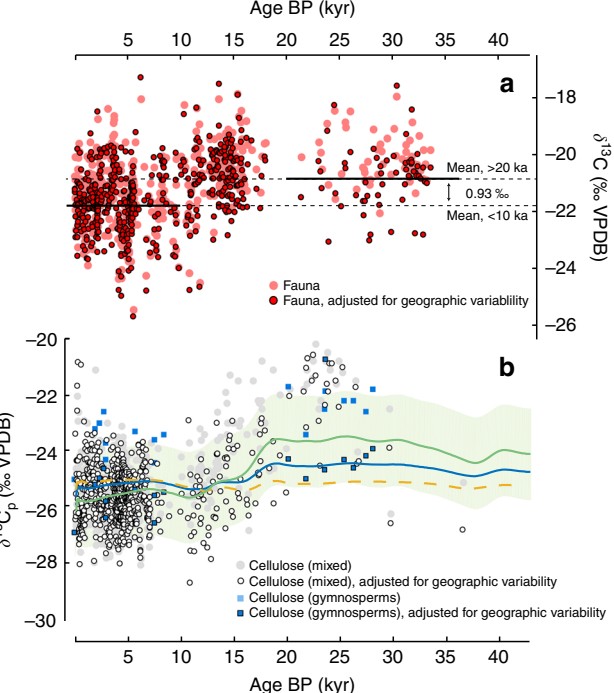

**Fig. 2** Stable carbon isotope composition of terrestrial fossil archives over the last deglaciation, from predominantly C$_3$ ecosystems. **a** Faunal collagen from northwestern Europe and northern Eurasia from 33.4 kyr BP to early 20$^{th}$ century. Light red circles show $\delta^{13}$C before adjustment for MAP, altitude and latitude. Dark red circles show adjusted $\delta^{13}$C, based on the regressions of Kohn[1]. Dark black lines indicate adjusted means of faunal $\delta^{13}$C before 20 kyr and after 10 kyr, which are −20.85‰ and −21.78‰, respectively. **b** Plant cellulose $\delta^{13}C_p$ records from the Northern Hemisphere, 40.5 kyr BP to early 20th century, with model curves. Colour codes match legend in Fig. 1: light green, SJ-2012; blue, Voelker-2016g; dashed yellow, Farquhar-1982. SJ-2012 model is shown with 1$\sigma$ propagated uncertainties. Dark blue squares show gymnosperm data adjusted for geographic variability in MAP, altitude, and latitude. Light blue squares show raw gymnosperm data before adjustment. Filled white circles and grey circles show mixed species (either angiosperm or unidentified) before and after adjustment for geographic variability, respectively

Our corrections for changes in MAP imply a residual pCO$_2$ effect during the deglacial rise in CO$_2$ (~80.5 ppmv) of −0.53‰ for fauna, or equivalently −0.7 ± 1.9‰ per 100 ppmv (Fig. 3a). Only gymnosperm species are represented across our entire plant cellulose compilation, and these species distinguish a larger pCO$_2$ effect of −1.7 ± 1.5‰ per 100 ppmv. Therefore, both collagen and cellulose records reflect changes in pCO$_2$ (in addition to changing MAP), but with different magnitudes. Our fauna primarily reflect a dietary contribution from angiosperms, whereas our plant compilation is biased towards gymnosperm species at the LGM. We suggest that the disagreement between our plant and faunal records is likely caused by a genuine physiological difference in leaf gas-exchange strategy between angiosperm and gymnosperm plants[31, 34].

The magnitudes of the pCO$_2$ effect implied by our data are consistent with models of photosynthesis which predict a dynamic leaf gas-exchange strategy, and a variable ratio of intercellular to atmospheric pCO$_2$ over the 180–400 ppmv range. This is less than that proposed by Breecker[16] for speleothems, −1.6 ± 0.3‰ per 100 ppmv (1$\sigma$), but greater than that proposed by Kohn[15] for fossil collagen, −0.03 ± 0.13‰ per 100 ppmv (2 s.e., see Fig. 3a). We suggest the latter discrepancy is due to the limited temporal resolution of that data set, which is neither large enough

nor sufficiently well dated to resolve millennial-scale shifts in $\delta^{13}C_p$.

With respect to our gymnosperm cellulose record, we find that $\delta^{13}$C (adjusted for geographical variability) is best described by SJ-2012 and Voelker-2016g (SJ-2012; RMSE = 1.07, AIC = 25, BIC = 31; Voelker-2016g; RMSE = 1.04, AIC = 26, BIC = 34). This is not surprising because SJ-2012 is biased towards gymnosperm palaeo-data below ~350 ppmv. Our faunal analysis shows that the SJ-2012 model leads to an overestimation of the pCO$_2$ effect for other plants and fauna, particularly at periods of low concentration. Therefore, although this model may be appropriate for gymnosperms, we suggest that SJ-2012 should not be used as a baseline to infer changes in angiosperm plants and hence the majority of ancient fauna. With respect to these records, the Voelker-2016a model best reproduces the magnitude of the deglacial shift observed in fauna (−0.53‰, Supplementary Table 4), but is offset from all cellulose records. The ~1.4‰ offset is probably related to our choice of $a$ and $b$ constants, and/or inaccuracies in the fitted relationship between $c_i/c_a$ and $c_a$. Another alternative is some hitherto unknown subtlety of the isotopic relationship between fauna and bulk diet. This last scenario is unlikely because the difference between the cellulose and collagen records, averaged over the Holocene, imply an average collagen-diet enrichment factor consistent with the value of 5.1‰ determined from modern controlled-feeding studies[35–37], after factoring in the ~1‰ isotopic offset between cellulose and bulk leaf tissue (faunal diet)[38]. Further chamber and palaeo-data from angiosperm plants, across a wider range of pCO$_2$, might be needed to shed light on the other explanations.

## Discussion

Faunal $\delta^{13}$C studies have been used to interpret changes in forest cover across Western Europe during the last deglaciation. For example, isotopic analysis of late Pleistocene roe deer in northern France show a range from −19.0 to −20.9‰, and a shift to values more negative than −22.5‰ during the Holocene has been used to infer the presence of a 'canopy effect' on faunal $\delta^{13}$C during the deglaciation, considering only a correction for past changes in $\delta^{13}C_{CO_2}$[39], if pCO$_2$ effects are negligible. Although similar interpretations have been challenged[28, 29, 40], presently the canopy effect and water availability are more frequently cited as the driving parameter behind $\delta^{13}$C trends of western European fauna during the late Pleistocene/Holocene transition[30, 41].

Given the pCO$_2$ effect displayed by our data, a cutoff of −22.5‰ would overestimate the extent of the canopy effect on faunal $\delta^{13}$C. Values more negative than −22.5‰ also reflect the postglacial rise in pCO$_2$, via its effect on carbon isotope fractionation in C$_3$ plants. The extent to which a genuine canopy effect is also reflected in our faunal record is difficult to determine. However, it is unlikely to be significant. First, there are no strong differences in $\delta^{13}$C across different species during the Holocene (species show similar means at approximately −21.7‰, when adjusted for geographical variability). The opposite result would be expected from a canopy effect, as only some of our species are forest-dwelling. Second, modern studies from temperate woodlands show limited effects on faunal $\delta^{13}$C, even when a canopy effect is present in vegetation[42]. However, the possibility of some small contribution from changing canopy cover is difficult to fully exclude, therefore our pCO$_2$ effect for fauna is a maximum estimate. Considering the multiple lines of evidence, including plant chamber experiments and the fossil record, it is now more plausible to believe that shifts in terrestrial $\delta^{13}$C during the last deglaciation primarily reflect changes in pCO$_2$, along with smaller contributions from changing MAP and possibly increased canopy

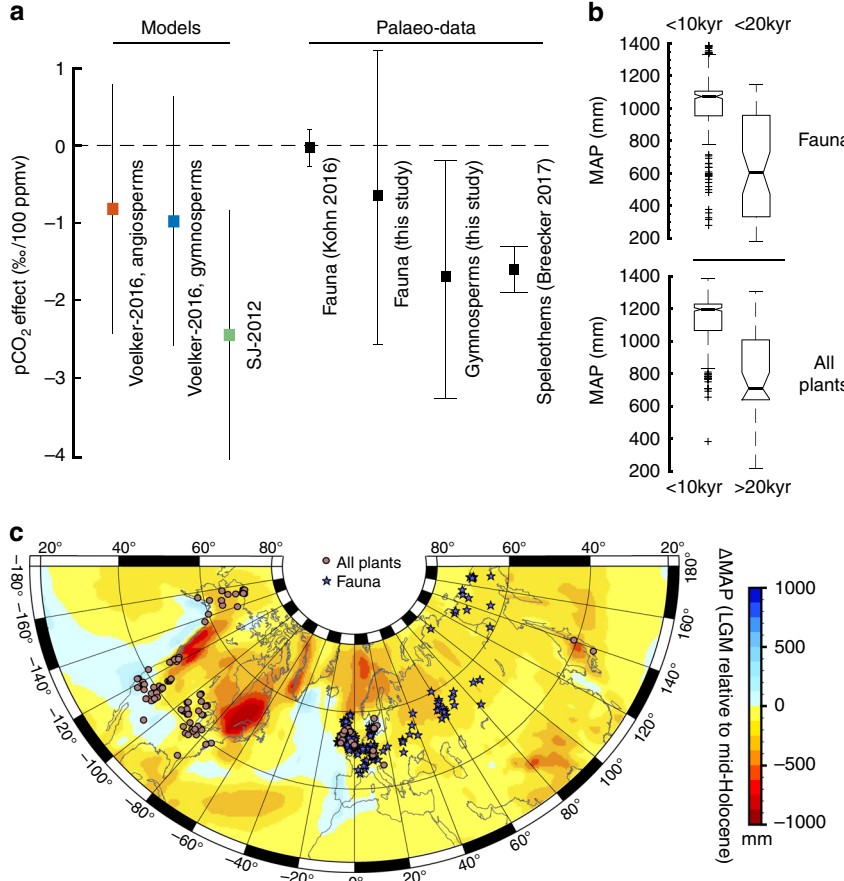

**Fig. 3** Effect of atmospheric $pCO_2$ on mean stable carbon isotope composition over the last deglaciation, determined from terrestrial records and from changes in MAP. **a** Magnitude of $pCO_2$ effect predicted by models and palaeo-data. Error bars for models, fauna, and gymnosperms (all this study) and speleothems[16] are $1\sigma$ propagated uncertainties, and 2 s.e. for fauna from Kohn[15]. **b** Changes in MAP predicted by PMIP3-CMIP5 multi-model ensemble between the LGM and mid-Holocene, used to constrain the $pCO_2$ effect on our data set. **c** Geographical distribution of our data, showing the magnitude of MAP changes predicted by the multi-model ensemble over the same time period

cover. Our finding is supported by analysis of ancient and modern $\delta^{13}C$ from wolves and bison bone collagen[43] which show strong correlations between $pCO_2$ and $\delta^{13}C$.

More generally, we propose that similarly rapid shifts in the carbon isotope baseline of $C_3$ plants need to be considered throughout the Quaternary, particularly during periods of low $pCO_2$, when atmospheric effects on photosynthetic fractionation are magnified. Faunal isotopes from predominantly $C_3$ and mixed $C_3/C_4$ environments are routinely used to infer regional ecologies, under the assumption that $\delta^{13}C$ mainly reflects growing season, mean annual temperature[44] and MAP. We urge caution in this approach, since our analysis shows that shifts in terrestrial archives may simply reflect rapidly changing atmospheric chemistry, and not other environmental variables. Beyond the Quaternary, there is also evidence to suggest that other atmospheric variables may lead to significant changes in the carbon isotope baseline in deep time. Plant chamber experiments have revealed strong relationships between carbon isotope discrimination and changing $pO_2$[34, 45], since Rubisco also as an affinity for oxygen. Whilst potentially relevant to geological periods of subambient or elevated $pO_2$, the influence of oxygen on plant $\delta^{13}C$ is probably minimal during the Quaternary because $pO_2$ levels remained relatively stable over much of this period[46]. The short-term shifts we observe on millennial timescales are therefore more likely due to changes in $pO_2$. In future, $pCO_2$- and possibly $O_2$-dependent models of carbon isotope fractionation should be used together with the regressions of Kohn[1] to

reconstruct changes in MAP or other atmospheric variables using ensembles of terrestrial archives. Rapid advancements in ice core measurements may help extend this approach further back in time, and offer more accurate tools for the reconstruction of ancient environments. Finally, there is now an intriguing body of evidence[31, 34] which reveals strong phylogenetic differences in the carbon isotope response of plants to atmospheric extrema, but it is clear that several questions remain about the precise nature of these relationships. There is an urgent need for improved comprehensive models of photosynthetic fractionation, which are essential for $pCO_2$ reconstruction beyond ice core records, as well as predicting uptake of future anthropogenic $CO_2$ emissions by the terrestrial biosphere[47].

## Methods

**Materials**. Plant $\delta^{13}C$ records were compiled from previously published studies of tree wood and *Pinus* needles, which were all pretreated to $\alpha$-cellulose, with the exception of more recent wood samples[48, 49], which were pretreated using standard acid–base–acid (ABA) protocols. Radiocarbon dates of collagen and plants were calibrated using OxCal v. 4.2[50] using the IntCal13 calibration curve[51] whenever raw radiocarbon determinations were reported, and reported in kyr BP (where BP = 1950 CE). Faunal collagen $\delta^{13}C$ was compiled from the Oxford Radiocarbon Accelerator database as well as other previously published sources, are presented in Supplementary Data 1 (see also Supplementary Fig. 2). We selected herbivore cellulose from predominantly $C_3$ environments (all >90%, most >99.5% according to ref. [52]), excluding Reindeer (*Rangifer tarandus*) and other species known to consume large amounts of lichen. Over half our record comprises grazing and browsing ungulates (Supplementary Fig. 3), e.g. *Cervus spp.* (19%), *Bos spp.* (24%) and *Equus spp.* (17%).

**Adjustments for geographic variability**. Collagen and cellulose $\delta^{13}C$ values were adjusted to MAP = 1000 mm, altitude = 840 m, latitude = 50 °N, using Eq. (1) of Kohn[1], and the following procedure. First we obtained the altitude (m) for each location according to the GTOPO30 digital elevation model, available at https://lta.cr.usgs.gov/GTOPO30 (Accessed 13 Nov 2016). We also calculated modern annual precipitation for each location according to the WorldClim v. 1.4 model[53], averaged over the period 1960-1990 AD[53]. We calculated adjusted values according to $\delta^{13}C_{adj} = \delta^{13}C + \delta'$, where $\delta' = \delta_{MAP} + \delta_{alt} + \delta_{lat}$. In turn, Eq. (1) of Kohn[1] yields $\delta_{MAP} = -5.6 \times \log_{10}(1000 + 300) - (-5.6 \times \log_{10}(MAP + 300))$, $\delta_{alt} = 0.00019 \times (840 - \text{altitude})$ and $\delta_{lat} = -0.0124 \times 50 - (-0.0124 \times |\text{latitude}|)$.

**Calculation of pCO₂ effect**. We calculate the magnitude of the $pCO_2$ effect across the LGM-Holocene transition according to the following assumption: $\Delta(\delta^{13}\overline{C}_{adj}) \approx \Delta(\delta^{13}\overline{C}_{pCO_2}) + \Delta(\delta^{13}\overline{C}_{MAP}) + \Delta(\delta^{13}\overline{C}_{CO_2})$. Here $\Delta(\delta^{13}\overline{C}_{adj})$ is the total difference in mean faunal or plant $\delta^{13}C$ adjusted for geographic variability in latitude, altitude and MAP, between >20 and <10 kyr. The latter values exclude the industrial era. Examination of the ice core records between >20 and <10 kyr shows that $\Delta(\delta^{13}\overline{C}_{CO_2}) \simeq 0.05$‰, which is negligible, therefore: $\Delta(\delta^{13}\overline{C}_{adj}) \approx \Delta(\delta^{13}\overline{C}_{pCO_2}) + \Delta(\delta^{13}\overline{C}_{MAP})$. This expression assumes that the total difference in adjusted $\delta^{13}C$ is approximately equal to the combined contributions of changing MAP, denoted as $\Delta(\delta^{13}\overline{C}_{MAP})$, and changing $pCO_2$, $\Delta(\delta^{13}\overline{C}_{pCO_2})$. By rearranging this expression, and subtracting the contribution from changing MAP, we evaluated the magnitude of residual changes due to changing $pCO_2$. $\Delta(\delta^{13}\overline{C}_{MAP})$ was obtained using $\Delta(\delta^{13}C_{MAP}) = -5.6 \times \log_{10}(MAP_{LGM} + 300) - (-5.6 \times \log_{10}(MAP_{mid-H} + 300))$. To estimate regional changes in MAP, and associated uncertainties, we utilised seven coupled general circulation models (GCMs) of MAP at the LGM (21 kyr) and mid-Holocene (6 kyr). We obtained Palaeoclimate Modelling (PMIP3) and Coupled Modelling Intercomparison Project (CMIP5)[54, 55] ensemble predictions of average yearly precipitation flux at each locality, available at https://pmip3.lsce.ipsl.fr (accessed June 2016). The change in MAP predicted by these models is shown in Supplementary Fig. 4, and the change in MAP predicted by the ensemble model is shown in Supplementary Fig. 5. These ensemble predictions were only available for the LGM and the mid-Holocene, so we corrected >20 kyr data using the LGM ensemble predictions, and <10 kyr data with the mid-Holocene predictions (Supplementary Fig. 6). We choose GCM ensemble predictions for two reasons: first, because they are largely independent of assumptions about stable carbon isotopes and water availability, which would create circularities in our approach, and second, because uncertainty estimates may be derived for MAP change. Additionally, studies have shown that using a multi-model ensemble mean is more accurate than choosing one particular model[54, 56, 57]. The seven GCMs are named in the Supplementary Information, and all model outputs are provided in Supplementary Data 1.

**Models of photosynthetic fractionation**. We computed four models as follows: model 1 (Farquhar-1982) was Eq. (1) with a constant $c_i/c_a$ ratio chosen of 0.6. Although it is unlikely that plants employ a strategy of constant $c_i/c_a$ under atmospheric extrema, this value was selected as a reasonable compromise for $\delta^{13}C_p$ since it reproduced the modern globally averaged $\delta^{13}C_p$ value from ref. [1] of −27.1‰. Model 2 (Voelker-2016g) was taken to be Eq. (1) modified by a linear dependence of $c_i/c_a$ on $c_a$ for gymnosperms given by ref. [31] as $c_i/c_a = 0.00038c_a + 0.502$. Model 3 (Voelker-2016a) was a similar model for woody angiosperms with $c_i/c_a = 0.00031c_a + 0.649$. For all three of these models, we used $a = 4.4$‰ based on arguments of the diffusivity of atmospheric $CO_2$, which is proportional to the square root of the reduced masses of the two isotopologues $^{13}CO_2$ and $^{12}CO_2$[58]. For $b$, we used a value of 28.2‰, which also reasonably reproduces modern globally averaged $\delta^{13}C_p$ using Voelker-2016g. It should be noted that Eq. (1) is a simplification of an expanded and refined equation given in ref. [4] that incorporates dissolution of $CO_2$ in solution, and diffusion inside the leaf, as well as discriminations associated with dark respiration and photorespiration. Surprisingly little discussion has been made as to the possible influence of the effects of photorespiration and dark respiration in deep time, which should also exhibit a strong dependence on $1/c_a$ (i.e. magnified at low $c_a$). These components are intrinsically difficult to measure in modern plants. We assumed both effects were negligible. We think omission is justified, since these terms imply a shift in the carbon isotope composition of terrestrial $C_3$ plants which is unreasonable in magnitude (~3–4‰), and in the wrong direction (i.e. a depletion in $^{13}C$ at the LGM relative to Holocene data) to account for the deglacial rise in $pCO_2$. Model 4 (SJ-2012) was the generalised hyperbolic model of Schubert and Jahren[13], based on Eq. (2), with constants taken from fits performed in that paper, which are $A = 28.26$, $B = 0.22$, and $C = 23.9$. For the curves for SJ-2012 presented in Figs. 1 and 2, error bounds represented an expanded uncertainty in $\delta^{13}C$, which is the $1\sigma$ range of the data set of Kohn[1] (1.62‰) combined in quadrature with $1\sigma$ propagated uncertainies from ice core splines. In other words, our error bounds considered both the range of the distribution of modern $\delta^{13}C$ values, and analytical error in ice core measurements. They are therefore conservative estimates that represent the plausible range of $\delta^{13}C$ values which may be expected in $C_3$ plants according to that model. The magnitude of error bounds was similar for all other models, and for simplicity error bounds for other models were not drawn in our figures, and we choose to draw those only for SJ-2012.

We compared the four models with both our cellulose record and our faunal data, shifted to the equivalent plant values by subtracting $\varepsilon^*_{d-c} = 5.1$‰. For model comparison we used three complimentary statistics: root mean square error (RMSE), Bayesian information criterion (BIC) and Akaike information criterion (AIC), all calculated from the residuals of each model fit to our records. The advantage of the latter two statistics is that they allow model selection based on a trade-off between goodness-of-fit and model complexity. Briefly, if predicted values of model $M$ at time $t$ are represented $y_t$ and data are represented by $y_i$, we calculated RMSE as $\sqrt{\left[\sum_{t=1}^{n}(y_t - y_i)^2\right]/n}$. BIC is calculated from $\ln(n)k - 2\ln(\mathcal{L})$, where $\mathcal{L}$ is the maximised value of the likelihood for mdoel $M$, with number of free parameters $k$. AIC is calculated as $2k - 2\ln(\mathcal{L})$. All goodness-of-fit statistics are shown in Supplementary Tables 1–3.

**Ice core compilation**. $CO_2$ concentrations were obtained from six Antarctic ice cores, EPICA Dome C (EDC)[59–64], Talos Dome[33, 65], EPICA Dronning Maud Land (EDML)[33, 64–67], Siple[68, 69], and records showing recent anthropogenic rise in $CO_2$ concentration from the Law Dome and South Pole[70]. Pre-industrial records were synchronised on the AICC2012 timescale[71]. We obtained a Monte Carlo smoothing spline according to the procedure outlined in ref. [60]. We performed 1000 replicate cubic spline fits to the entire data series, with input data picked randomly from the $1\sigma$ error range, and applied a 375 yr cutoff period to exclude high-frequency noise. Records for the anthropogenic era from the Law Dome and South Pole are presented on the age scale of ref. [70], cubic splined without a cutoff period, and then combined with the pre-industrial curve. The curve obtained for the corresponding pre-industrial $\delta^{13}C_{CO_2}$ records was taken from ref. [33] and was obtained by a similar procedure, synchronised on the AICC2012 timescale, as well as incorporating records from the Law Dome and South Pole.

**Data availability**. Model output and all data used in the current study are made available in the figshare repository, 10.6084/m9.figshare.5497918. Additionally, radiocarbon dates may be queried using the OxCal database, where appropriate: https://c14.arch.ox.ac.uk/database/db.php?page=oxaResult.

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

## Acknowledgements

We thank Sarah Eggleston, Ben Hmiel, Lee Murray, and Vinesh Rajpaul for their advice, and Patrick Roberts for his suggestions and critical reading. The manuscript was much improved by the constructive comments of the anonymous reviewers. V.J.H. and E.L. received support from the Clarendon Fund, University of Oxford. We acknowledge support from the UK Natural Environment Research Council (NERC) for the Oxford node of the national NERC Radiocarbon facility. Open access fees were kindly provided by Oxford University's RCUK Open Access Block Grant. We also acknowledge the World Climate Research Programme's Working Group on Coupled Modelling, which is responsible for CMIP, and we thank the climate modelling groups for producing and making available their model output.

## Author contributions

V.J.H. designed research, E.L., A.J. and C.B.R. assisted research. V.H. analysed the data and wrote the paper with input from all authors.

## Additional information

**Competing interests:** The authors declare no competing financial interests.

