## [Peer review file · Nature Communications]

Reviewers' comments:

Reviewer #1 (Remarks to the Author):

Hare and colleagues compile records of plant $\delta^{13}\text{C}$, animal $\delta^{13}\text{C}$, atmospheric $\delta^{13}\text{C}$, and atmospheric CO_2 concentration for the last glacial-interglacial transition. They then apply four plant physiological models in order to judge if the effects of CO_2 concentration on plant $\delta^{13}\text{C}$ leave a resolvable signal in natural archives. This study has implications for many in the earth and biological sciences (and the authors summarize these implications), and so may be appropriate for publication in Nature Communications.

I have one general comment that I do think requires serious consideration by the authors. As with any natural experiment, one needs to fully explore all of the possible confounding factors. The authors explore the effects of mean annual precipitation (MAP), and report that a MAP doubling would be needed to explain their plant $\delta^{13}\text{C}$ shift. Because this change is "very large" (lines 108-109), the authors discount this possibility, but in doing so discount the effect of *any* MAP change. This doesn't seem prudent: the most likely scenario is that the plant $\delta^{13}\text{C}$ shift is affected by both rising MAP and rising CO_2 . Published climate models could give the authors predictions of MAP change for the regions where most of the compiled data come from.

Similarly, the authors discuss the potential effects of a closing canopy, but use a weak argument to exclude this possibility (lines 152-154). Again, a likely scenario is that the plant $\delta^{13}\text{C}$ shift is affected by both a closing canopy and rising CO_2 (and rising MAP). Published vegetation models may give insight into this effect, especially given that most of the compiled data come from regions whose vegetation shifted from tundra to forest.

A third related point is that the authors note that most of their glacial record is dominated by gymnosperm vegetation (line 126), while most of the Holocene record is dominated by angiosperms (line 131). This alone could explain the authors' observed shift in plant $\delta^{13}\text{C}$ to lower values. This is a critical point that I don't think the authors address.

Minor comments:

Title: may be misleading for folks who work with radiocarbon.

Line 106: residual effect for collagen is 0.5 per mil, not 1 per mil.

Figure 1 caption: say that the white numbers represent the plant $\delta^{13}\text{C}$. Also, are the white dots estimates every X years?

Reviewer #2 (Remarks to the Author):

Review of Hare et al.

Hare et al. present new carbon isotope values of plants and fauna over the past ~ 30 ka. This manuscript specifically addresses and indicates that it is going to test the hotly debated effect of $p\text{CO}_2$ on plant carbon isotopes. This problem has been debated since the 1960s work of Epstein. Much work has been done to address this issue, but one of the key challenges may be how the studies are setup (field-based vs growth chamber), timescales (annual, decadal, centennial, etc), and unconstrained variables (atmospheric $^{13}\text{CO}_2$, water, vegetation, etc). This manuscript claims that it is going to solve this problem by using plant and faunal $\delta^{13}\text{C}$ values from the fossil record, spanning the past ~ 30 ka, to compare different $p\text{CO}_2$ plant fractionation models. On the surface, this should be exciting.

However, this study is not setup in a way to test the effects of pCO₂ on plant δ¹³C values. There are fatal flaws in approach that preclude the authors from making almost all of the arguments presented here. As someone who works in these areas, it is shocking to see that this study makes many of the same mistakes as other studies, that good work is thrown completely discounted (effect of water on plant δ¹³C), that major changes in vegetation and precipitation from ~30 ka to present are completely ignored (this is MAJOR), and that although this study discounts water, it actually controls for water over the past ~30ka, but with modern MAP values! There are many other major mistakes in the setup, logic, and argument here.

This is such a fabulous dataset, I encourage the authors to think about how they could better use this study to learn about the past and to make a more novel contribution than testing the importance of pCO₂ and MAP with poor constraints. Isn't there something much more interesting that could be extracted about paleoclimate and paleovegetation given that you have fauna and flora from 30ka to present?

Significant problems:

Using plant and faunal δ¹³C values through time: This study does a poor job controlling for vegetation changes through time and more importantly, the effect of water. Water influences plant δ¹³C values (See below). There are MAJOR changes in water over the past 30 ka! Adjusting values to a 1000 mm precipitation value, based on MODERN values, is not justifiable and makes absolutely no scientific sense (line 102). Constrains on water through time must be made and there is no way this can be done in this study correctly. We just do not have good estimates of paleo precipitation over this time period to control for precipitation to evaluate changes in pCO₂, especially when sampling varies spatially and temporally (Supp figure 4). But there is good evidence of major widespread dryness associated with glaciations and the LGM, and into the Holocene. To make matters worse, we don't even have good agreement between proxies and climate model outputs. Without a careful control on spatial and temporal changes in precipitation, there is absolutely no way to use the data generated here to test for the effects of pCO₂ on plant carbon isotopes, and by extension, the fauna.

Line 108: "the residual shift of 1 ‰ would imply an approximate doubling of global MAP from the LGM to the Holocene". There is no way that a 1‰ change can be interpreted as a doubling of MAP as an argument here! A doubling of precipitation is not unlikely at this time. If precipitation is low (a few hundred mms), then doubling it is not unlikely. But, this change of 1‰ could also be caused by biases in the spatial coverage of samples, especially at this time period when there are major changes in climate and vegetation! The effect of MAP on δ¹³C is also not linear, and therefore requires knowing MAP before or after the change. A minor point, this sentence refers to global MAP, but the study is restricted to North America and Europe.

This study suggests that mean annual precipitation (MAP) has been argued to be the most important control on plant δ¹³C values (Kohn, Diefendorf et al). This is not the case. Both studies find strong relationships between MAP and δ¹³C, but these are not the only controls and both studies are clear about that. Vegetation, plant functional type, altitude, and many other variables are important controls, but combined, these still do not explain all of the variation in plant δ¹³C. This study cannot test the influence of water (see next issue) and it is completely inappropriate to suggest that water is not important based on this study (see abstract).

Line 19-21: This rationale/setup of the problem is completely wrong here. As written, the manuscript suggests that Kohn (2016) disputes the Breecker et al (2016) study and argues that the speleothem record could be explained by MAP alone. This is just not true. The Kohn study came out before the Breecker study and therefore makes no mention of it. The Kohn study focuses on earlier time periods and at no point argues that MAP is the only effect on faunal (or plant for that matter) δ¹³C values. There are arguments about the effect of pCO₂ on plant δ¹³C values, but

there is no argument in the literature that MAP influences plant $\delta^{13}\text{C}$ values, at least to some degree, but MAP it is not the ONLY control. This argument here and throughout the manuscript is just wrong.

Lines 103-105: You are not controlling for water here. MAP of today is not the MAP of the past for Europe or North America, especially over the past ~ 30 ka!!!!

Lines 115-118: The argument that you can discount the Kohn 2016 study because of temporal resolution is bogus and this study is NOT setup in a way to validate the Kohn study. There are major differences in the approach between these different studies (including this one). Kohn presents data on million year timescales, so the rationale to throw it out because of poor dating and that it does not capture millennial resolution is just wrong. There are also many other geologic studies that do not find any relationship between $p\text{CO}_2$ and plant/faunal $\delta^{13}\text{C}$ studies.

Lines 122-140: If the collagen-diet enrichment is going to be tested here, then more unknowns need to be controlled for. Water and vegetation ("mostly angiosperm" and mostly C3) both need to be accounted to test this and not enough information is provided here.

Lines 144-172: This is all vague and not well supported. The examples are not explained well enough and not enough information is provided to get a sense if the arguments are justified.

Minor comments

Line 181 "selected herbivore cellulose". Collagen?

Figure 1 is not useful.

Figure 2: Putting the modern mean plant $\delta^{13}\text{C}$ value on this figure is misleading and suggests there is one mean value through time. Obviously that is not the case. This also ignores atmospheric $\delta^{13}\text{C}$ changes.

Reviewer #3 (Remarks to the Author):

General Summary:

This manuscript addresses a currently relevant topic about the effect of carbon dioxide concentration on the isotopic value of plant material in the fossil record. The debate over this topic spans several decades, but has become prominent because of relatively recent findings that argue strongly both for dependence and for independence of the carbon isotopic value of plants with changing CO_2 concentration. Finding the answer to the question of dependence/independence will be important, because the carbon isotope value of plants is currently being used to infer diet and habitat changes in animal records, but also is used as part of methods that infer CO_2 in deep time. The latter case (^{13}C in models of CO_2 estimation) would create circularity and hold back efforts to produce long-term CO_2 records that are compared to records of temperature. Thus, any manuscript that provides insight into the dependence or independence of ^{13}C of plant material (such as the one here) will be of interest to a wide swath of the deep time paleoanthropological, paleontological, and paleoclimate communities.

The authors utilize a recent high-resolution compilation of carbon dioxide concentrations (and the isotopic value of that atmospheric carbon dioxide) from a suite of ice cores, stretching back 155ka.

They use this data to compute predictions of the ^{13}C values of plant organic material using four models of photosynthesis to identify trends in the data for the past 155ka. With two high-resolution datasets for the past $\sim 35\text{ka}$, they compare how their model fits with the plant and animal data sets, to determine the degree of the effect of atmospheric CO_2 concentration on these two carbon isotope datasets. They use these results to conclude that there is an $\sim 1\text{‰}$ effect on the isotopic value of plant material per 100 ppm of CO_2 change. This value that is less than that found for speleothem records ($1.66\text{‰}/100\text{ ppm}$; Brecker, 2017) and for experimental results on radishes and Arabidopsis plants (up to $2\text{‰}/100\text{ ppm}$; Schubert and Jahren 2012). This is in contrast to a different set of carbon isotopic records for the same interval from plants and animals (Kohn 2016) that suggest that ^{13}C of plants is effectively independent of CO_2 concentration ($0.0 \pm 0.3\text{‰}/100\text{ ppmv}$).

General Comments:

The manuscript is well written and the figures are well crafted and very clear. The literature is well cited and up-to-date, and the argument is clearly stated. The authors don't stretch their conclusions beyond the data that they present. Their data is nicely documented in the text and supplemental information.

The novel approach taken here is to determine if there is a dependence of plant ^{13}C on the concentration of CO_2 , by attempting to understand the effect to $p\text{CO}_2$ through models of plant photosynthesis. They use four photosynthesis models (two new, one from Schubert and Jahren, and one modified from Farquhar) to compare their predictions of plant ^{13}C (based upon ice-core data) to standardized values of faunal and plant isotopic data. Commendably, they go to great lengths to standardize their data to the same geographic parameters, which helps their argument. Looking at the data in Figure 3 for fauna I'm fairly convinced that there is a 1‰ shift during the LGM for collagen, and somewhat also convinced for the cellulose record, though there is a large degree of scatter.

However, to my eye, it is the models that seem questionable. None of them are a very good fit to the sample data, so they fall a little short towards supporting the viewpoint that $p\text{CO}_2$ levels are the cause of the $\sim 1\text{‰}$ shift in $^{13}\text{C}_p$. Yes, all models are capable of modeling the anthropogenic influence on $^{13}\text{C}_p$ (except the modified Farquhar 1982 model), but not all of the models are capable of correctly predicting the 1 per mil shift (line 124) in plants. The SJ12 doesn't match the 1 per mil shift (line 124) in plants. The model based upon gymnosperms does match the 1‰ shift, but it fits more poorly for the interval when gymnosperms were the major component of the vegetation (pre-Holocene), and fits better for the part of the record dominated by angiosperms. The angiosperm model has the worst fit to the plant cellulose record, and doesn't even overlap with the part of the record that is dominantly from angiosperm tissue. The $p\text{CO}_2$ independent model (modification of Farquhar) fits the data to a certain degree (though not as well as the SJ12 model) but doesn't match their conclusions of a $1\text{‰} / 100\text{ ppm}$ effect of $p\text{CO}_2$ on $^{13}\text{C}_p$. From their model results, there may be a small effect of $p\text{CO}_2$ levels, but there still remains a significant portion of the data that is not explained by any of the four models.

Thus, some other environmental factor, or parameter in the model must be considered before we can use this study to finalize the debate over the effect of $p\text{CO}_2$ on plant carbon isotopic values. Also, there is an insufficient discussion of the changes in source carbon (reservoirs) that occurs during the LGM transition to the Holocene. For instance, I would like to see a comparison of the methane record from the same ice-cores, as this has a much lighter isotopic composition than carbon dioxide, and could explain a considerable amount of the shift towards lighter isotopic composition of the atmospheric CO_2 and therefore the isotopic value of the plants and then the collagen. If the effect of the isotopic reservoir of the carbon is negligible, then it should be discussed in greater detail and is largely lacking in this manuscript.

The section on "Prediction of plant isotopes over the last glacial cycle" needs further explanation for us to understand how it has much relevance to the debate over the effect of pCO₂ on $\delta^{13}\text{C}_{\text{p}}$, considering that their comparative sample records don't cover the entire interval of 155ka. They identify three intervals of rapid $\delta^{13}\text{C}_{\text{p}}$ change (12–18, 60–62.7 and 129.4–135 ka), but they only have sample data for the interval of 12-18ka. This section does create a useful set of hypotheses to test, but is only useful for later comparison with high-resolution records (like they have for the LGM) over the 155ka interval. Also, it isn't adequately explained why a change of 0.25‰/ka is considered an important rate. It may be, but as written, this section seems like an output from the modeled ice-core data, rather than supporting evidence for their argument for a dependence of $\delta^{13}\text{C}_{\text{plant}}$ on pCO₂.

AUTHORS' RESPONSE TO REVIEWERS: NCOMMS-17-08406-T:
"ATMOSPHERIC CO₂ CONTROL ON FOSSIL CARBON ISOTOPES"

General

We thank the editor and the reviewers for their replies, and the prompt review process. We agree with several of the constructive criticisms raised by the reviewers, and are encouraged that all three reviewers consider that our dataset has the potential to shed light on a significant and topical issue for many in the earth and biological sciences. We have therefore incorporated significant new analyses of the dataset, which we believe have improved our arguments and the impact of the manuscript. In particular, we have now included a detailed sensitivity analysis of precipitation change over the LGM/Holocene transition for all of our data, which we base on an ensemble of published climate models. This analysis allows us to obtain bounds on the pCO₂ effect, along with uncertainties, which we compare with previously published estimates. Important differences also emerge between angiosperms and gymnosperms. Along with several other significant edits, we trust that these improvements will provide suitable grounds for further review.

Reviewer #1

Hare and colleagues compile records of plant d13C, animal d13C, atmospheric d13C, and atmospheric CO₂ concentration for the last glacial-interglacial transition. They then apply four plant physiological models in order to judge if the effects of CO₂ concentration on plant d13C leave a resolvable signal in natural archives. This study has implications for many in the earth and biological sciences (and the authors summarize these implications), and so may be appropriate for publication in Nature Communications.

*I have one general comment that I do think requires serious consideration by the authors. As with any natural experiment, one needs to fully explore all of the possible confounding factors. The authors explore the effects of mean annual precipitation (MAP), and report that a MAP doubling would be needed to explain their plant d13C shift. Because this change is "very large" (lines 108-109), the authors discount this possibility, but in doing so discount the effect of *any* MAP change. This doesn't seem prudent: the most likely scenario is that the plant d13C shift is affected by both rising MAP and rising CO₂. Published climate models could give the authors predictions of MAP change for the regions where most of the compiled data come from.*

We fully agree that plant d13C is affected by both rising MAP and rising CO₂. We have now incorporated an analysis of MAP change at each locality using a multi-model ensemble mean of PMIP3-CMIP5 climate models, which we discuss in greater detail in the revised SI, and in the manuscript main text (see new Fig. 3). The analysis has also allowed us to set better constraints on the pCO₂ effect, and key differences emerge between angiosperm and gymnosperm plants.

Similarly, the authors discuss the potential effects of a closing canopy, but use a weak argument to exclude this possibility (lines 152-154). Again, a likely scenario is that the plant d13C shift is affected by both a closing canopy and rising CO₂ (and rising MAP). Published vegetation models may give insight into this effect, especially given that most of the compiled data come from regions whose vegetation shifted from tundra to forest.

We agree that we did not develop the arguments about the canopy effect as fully as possible. The section “Implications for terrestrial carbon archives and MAP reconstruction” (lines 158 onwards) has been revised to discuss this in greater detail. Although it might be present, we do not believe that the canopy effect is significant in our datasets, for a number of reasons. There are no strong differences in $\delta^{13}\text{C}$ across different species during the Holocene, e.g. *Bos spp.*, *Cervus spp.* and *Equus spp.* all show similar means at ~ -21.7 per mil, when adjusted for geographical variability. The opposite result would be expected from a canopy effect, where we would expect species with different feeding strategies to reflect clear isotopic differences. Second, modern studies from temperate woodlands show limited effects on faunal $\delta^{13}\text{C}$, even when a canopy effect is present in vegetation (see ref [42] in main text). Even in tropical forests, there is a wide range of variability, and old leaves have been shown to exhibit a vertical canopy effect, whereas young leaves, shoots and flowers do not (see Roberts *et al.*, 2017, *Stable carbon, oxygen, and nitrogen, isotope analysis of plants from a South Asian tropical forest: Implications for primatology*, *Am. J. Primatology*). As it stands, our interpretation of the magnitude(s) of the $p\text{CO}_2$ effect provide a framework which consistent with several independent strands of evidence, and without any unambiguous evidence for a canopy effect it is difficult to argue for another point of view. Potentially there is a lot more to be discussed here, but unfortunately it is beyond the scope of the present manuscript.

A third related point is that the authors note that most of their glacial record is dominated by gymnosperm vegetation (line 126), while most of the Holocene record is dominated by angiosperms (line 131). This alone could explain the authors’ observed shift in plant $\delta^{13}\text{C}$ to lower values. This is a critical point that I don’t think the authors address.

This is an extremely important point, and we thank the reviewer for pointing it out. We have now addressed the issue in further detail (discussion in lines 124-130, 140-143; Fig. 2 updated to reflect gymnosperms; new Fig. 3 comparing our estimates for $p\text{CO}_2$ effect in gymnosperms and fauna with previously published data). Since gymnosperms species are represented across our record, we estimate the magnitude of the $p\text{CO}_2$ effect separately for fauna and gymnosperms. The effect for fauna is less than for gymnosperms, which is not surprising considering that they reflect an angiosperm dietary contribution, and other studies have found differences in $p\text{CO}_2$ between angiosperms and gymnosperms. The differences are now discussed, and compared with models that show different behaviours in these plant groups.

Title: may be misleading for folks who work with radiocarbon.

Yes, we agree. The effect of $p\text{CO}_2$ on radiocarbon is negligible, and we do not discuss it here. We have now changed it to refer to stable carbon isotopes.

Line 106: residual effect for collagen is 0.5 per mil, not 1 per mil.

Thank you. This section has been revised and updated to reflect the fact that the residual effect is different for collagen and cellulose (lines 112 onwards), which is actually a crucial point. We have also included a discussion (127 onwards) of the probable reasons for this disagreement (see also comments above to your third point).

Figure 1 caption: say that the white numbers represent the plant $\delta^{13}\text{C}$. Also, are the white dots estimates every X years?

We have now included this in the figure caption. The white dots are estimates every 20 years, from 1950 onwards. Before 1950 they represent 1000 year intervals. Note that on the advice of Reviewer 2, and to allow space for another figure showing the $p\text{CO}_2$ effect and MAP corrections, we have moved this figure to the supplementary materials and shortened this section.

Reviewer #2

Hare et al. present new carbon isotope values of plants and fauna over the past ~ 30 ka. This manuscript specifically addresses and indicates that it is going to test the hotly debated effect of

pCO₂ on plant carbon isotopes. This problem has been debated since the 1960s work of Epstein. Much work has been done to address this issue, but one of the key challenges may be how the studies are setup (field-based vs growth chamber), timescales (annual, decadal, centennial, etc), and unconstrained variables (atmospheric ¹³C, water, vegetation, etc). This manuscript claims that it is going to solve this problem by using plant and faunal $\delta^{13}\text{C}$ values from the fossil record, spanning the past ~30 ka, to compare different pCO₂ plant fractionation models. On the surface, this should be exciting.

However, this study is not setup in a way to test the effects of pCO₂ on plant $\delta^{13}\text{C}$ values. There are fatal flaws in approach that preclude the authors from making almost all of the arguments presented here. As someone who works in these areas, it is shocking to see that this study makes many of the same mistakes as other studies, that good work is thrown completely discounted (effect of water on plant $\delta^{13}\text{C}$), that major changes in vegetation and precipitation from ~30 ka to present are completely ignored (this is MAJOR), and that although this study discounts water, it actually controls for water over the past ~30ka, but with modern MAP values! There are many other major mistakes in the setup, logic, and argument here.

The reviewer's observation is correct that we did not attempt to control for *changes* in MAP across the deglaciation, which meant that our original corrections could only set a maximum bound on the pCO₂ effect. We did this precisely because it is difficult to place constraints on MAP change across the period. We acknowledge that this is one aspect which could be improved, and have now included corrections for MAP change based on an ensemble mean of 7 coupled general circulation models. There is a body of palaeoclimate literature (see SI for discussion) which suggests that an ensemble mean is the most accurate method of inferring palaeoclimate variables. Our corrections are generally larger than those applied in the study of Breecker (2017), who bases his corrections on proxy evidence, and are therefore more conservative. Our analysis now allows us to estimate the magnitude(s) of the pCO₂ effect from our data, which notably agree with the predictions of some published models, and reveal key differences between angiosperms and gymnosperms, which has also recently been found by other studies.

A significant point of our paper was to avoid corrections which are based on assumptions about MAP change over the LGM/Holocene transition, many of which introduce circularities because they are in fact based on assumptions about carbon isotope proxies and water, which are almost impossible to test. In our original paper we performed a careful and conservative adjustment for MAP/latitude/altitude because we wished to remove strong geographic and vegetation biases, avoiding circularities. The adjustment was not originally designed to control for *changing* MAP over the LGM/Holocene transition, but rather to control for modern spatial variability. Without constraints on changing MAP, we were only able to place a maximum bound on the pCO₂ effect. To our knowledge, nobody has attempted to remove geographical variability before, and we believe it is an important step in discerning the residual global signature of changing atmospheric chemistry. We believe the approach gives us greater confidence in our data, which is echoed by Reviewer #3 ("*Commendably, they go to great lengths to standardize their data to the same geographic parameters, which helps their argument*"). Although we pointed out that the adjustment was to a *common* MAP, alt, lat, we believe it could have been clearer that the procedure was designed to reduce geographical variability. Note too that the adjustments for modern MAP cancel out when considering changes through time.

This is such a fabulous dataset, I encourage the authors to think about how they could better use this study to learn about the past and to make a more novel contribution than testing the importance of pCO₂ and MAP with poor constraints. Isn't there something much more interesting that could be extracted about paleoclimate and paleovegetation given that you have fauna and flora from 30ka to present?

Indeed, we could have extracted something about paleoclimate and paleovegetation from our dataset, but this would have to be based on the very assumptions about MAP/pCO₂ which we are trying to test, which are currently poorly understood. We believe that constraining the pCO₂ effect is an absolutely crucial piece of the puzzle, which will place future studies on much firmer footing.

Significant problems:

Using plant and faunal $\delta^{13}\text{C}$ values through time: This study does a poor job controlling for vegetation changes through time and more importantly, the effect of water. Water influences plant $\delta^{13}\text{C}$ values (See below). There are MAJOR changes in water over the past 30 ka! Adjusting values to a 1000 mm precipitation value, based on MODERN values, is not justifiable and makes absolutely no scientific sense (line 102). Constrains on water through time must be made and there is no way this can be done in this study correctly. We just do not have good estimates of paleo precipitation over this time period to control for precipitation to evaluate changes in pCO₂, especially when sampling varies spatially and temporally (Supp figure 4). But there is good evidence of major widespread dryness associated with glaciations and the LGM, and into the Holocene. To make matters worse, we don't even have good agreement between proxies and climate model outputs. Without a careful control on spatial and temporal changes in precipitation, there is absolutely no way to use the data generated here to test for the effects of pCO₂ on plant carbon isotopes, and by extension, the fauna.

We have now incorporated an analysis of MAP change at each locality using a multi-model ensemble mean of PMIP3-CMIP5 climate models, which we discuss in greater detail in the revised SI, and in the manuscript main text (see new Fig. 3). Whilst we agree that we do not have good estimates of palaeo precipitation during this time, we believe that using an ensemble mean of these climate models provides more careful control on temporal changes in precipitation than alternative methods. Using an ensemble of these models provides several advantages, notably that uncertainty estimates can be placed on changing MAP for each locality. The mean uncertainty is propagated into our estimates of pCO₂ effect, something which previous studies have not properly considered. It is also almost impossible, given the size of our dataset, and the complexities of proxy disagreement, to consider another method of correcting this dataset, e.g. as Breecker has done with speleothems using site palaeoenvironmental evidence. Climate models provide the most parsimonious correction.

Line 108: "the residual shift of 1 ‰ would imply an approximate doubling of global MAP from the LGM to the Holocene". There is no way that a 1‰ change can be interpreted as a doubling of MAP as an argument here! A doubling of precipitation is not unlikely at this time. If precipitation is low (a few hundred mms), then doubling it is not unlikely. But, this change of 1‰ could also be caused by biases in the spatial coverage of samples, especially at this time period when there are major changes in climate and vegetation! The effect of MAP on $\delta^{13}\text{C}$ is also not linear, and therefore requires knowing MAP before or after the change. A minor point, this sentence refers to global MAP, but the study is restricted to North America and Europe.

Climate models display wide variability in MAP change, but the ensemble mean shows that the average Holocene MAP is slightly less than double that of LGM map (see SI Fig. S5, S6 for histograms of MAP change for our entire dataset, and main text Fig. 3 for boxplots of MAP change). However, in some instances (notably North America) the opposite change is noted, with a fair degree of confidence. Indeed, the effect of MAP is non-linear – and we correct using the Kohn equation.

This study suggests that mean annual precipitation (MAP) has been argued to be the most important control on plant $\delta^{13}\text{C}$ values (Kohn, Diefendorf et al). This is not the case. Both studies find strong relationships between MAP and $\delta^{13}\text{C}$, but these are not the only controls and both studies are clear about that. Vegetation, plant functional type, altitude, and many other variables

are important controls, but combined, these still do not explain all of the variation in plant $\delta^{13}\text{C}$. This study cannot test the influence of water (see next issue) and it is completely inappropriate to suggest that water is not important based on this study (see abstract).

We do not argue that MAP is the only control; but we argue that it is the strongest variable (in modern studies), which is fairly clear from the regressions of Kohn, Diefendorf et al.

For clarity, in several instances we have now added text to the revised manuscript. For example (lines 5-6): “although correlations exist with variables that include plant functional type and altitude, $\delta^{13}\text{C}_\text{p}$ is most strongly correlated with mean annual precipitation (MAP).”

Line 19-21: This rationale/setup of the problem is completely wrong here. As written, the manuscript suggests that Kohn (2016) disputes the Breecker et al (2016) study and argues that the speleothem record could be explained by MAP alone. This is just not true. The Kohn study came out before the Breecker study and therefore makes no mention of it. The Kohn study focuses on earlier time periods and at no point argues that MAP is the only effect on faunal (or plant for that matter) $\delta^{13}\text{C}$ values. There are arguments about the effect of $p\text{CO}_2$ on plant $\delta^{13}\text{C}$ values, but there is no argument in the literature that MAP influences plant $\delta^{13}\text{C}$ values, at least to some degree, but MAP it is not the ONLY control. This argument here and throughout the manuscript is just wrong.

We have edited this section to clear up any ambiguity about the relationship between the Kohn and the Breecker study, e.g. lines 19-24 “This [SJ-2012] model has been disputed (Kohn, 2016) on the basis that (1) the change in $\delta^{13}\text{C}_\text{p}$ can be explained by an increase in MAP, differential organic degradation, and changes in $\delta^{13}\text{C}_\text{CO}_2$ and (2) that fossil collagen and tooth enamel from the Eocene to the historical era apparently do not discern a $p\text{CO}_2$ effect. On the other hand, globally-averaged records of speleothem ^{13}C appear to support a strong $p\text{CO}_2$ dependence over the past 90 kyr (Breecker, 2017), and the issue is thus unresolved.”

Lines 103-105: You are not controlling for water here. MAP of today is not the MAP of the past for Europe or North America, especially over the past ~30 ka!!!!

(Please see above comments regarding corrections for changing MAP, and changes to the manuscript)

Lines 115-118: The argument that you can discount the Kohn 2016 study because of temporal resolution is bogus and this study is NOT setup in a way to validate the Kohn study. There are major differences in the approach between these different studies (including this one). Kohn presents data on million year timescales, so the rationale to throw it out because of poor dating and that it does not capture millennial resolution is just wrong. There are also many other geologic studies that do not find any relationship between $p\text{CO}_2$ and plant/faunal $\delta^{13}\text{C}$ studies.

We disagree with the idea that millennial-scale shifts in $p\text{CO}_2$ can be properly identified with data which have dating resolution of several thousand years (as is the case with the Kohn faunal data from the Eocene to the present), and this is a key point of our paper. In lines 92-102 and lines 135-137 we address this directly, e.g.: “considering that high amplitude changes in $\delta^{13}\text{C}_\text{p}$ are predicted to occur during relatively brief periods (i.e. ~ 2.7-5.6 kyr), and faunal data from previous studies are thinly represented across several million years, it is unlikely that they provide the necessary temporal resolution to discern a possible $p\text{CO}_2$ effect. In other words, beyond the limit of radiocarbon dating (~50 kyr), fossil archives will have minimum age uncertainties of several thousand years, which makes evaluation of the $p\text{CO}_2$ effect impossible.

Lines 122-140: If the collagen-diet enrichment is going to be tested here, then more unknowns need to be controlled for. Water and vegetation (“mostly angiosperm” and mostly C_3) both need to be accounted to test this and not enough information is provided here.

The collagen-diet enrichment factor is not tested here. This value is reasonably well constrained by several studies, e.g. see table 2 of Drucker et al. (2008) *Palaeo-3*.

Lines 144-172: This is all vague and not well supported. The examples are not explained well enough and not enough information is provided to get a sense if the arguments are justified.
We have now revised this paragraph, and discussed the possibility of a canopy effect in greater detail.

Minor comments

Line 181 “selected herbivore cellulose”. Collagen?

Yes, thank you. This has now been corrected.

Figure 1 is not useful.

We agree that it is less useful in the main text, although we believe the contour plots are still a valuable illustration of the model predictions. We have therefore moved Figure 1 into the supplementary materials, and condensed the two sections “Models of photosynthetic fractionation in C₃ land plants” and “prediction of plant isotopes over the last glacial cycle” into one section. This has allowed us space later in the manuscript to discuss environmental effects in greater detail, and to show our estimate of the pCO₂ effect, constrained by general circulation models of MAP change over the last deglaciation.

Figure 2: Putting the modern mean plant $\delta^{13}\text{C}$ value on this figure is misleading and suggests there is one mean value through time. Obviously that is not the case. This also ignores atmospheric $\delta^{13}\text{C}$ changes.

We have edited the figure to remove the ambiguity. Note that we do factor atmospheric $\delta^{13}\text{C}$ changes into our models.

Reviewer #3

This manuscript addresses a currently relevant topic about the effect of carbon dioxide concentration on the isotopic value of plant material in the fossil record. The debate over this topic spans several decades, but has become prominent because of relatively recent findings that argue strongly both for dependence and for independence of the carbon isotopic value of plants with changing CO₂ concentration. Finding the answer to the question of dependence/independence will be important, because the carbon isotope value of plants is currently being used to infer diet and habitat changes in animal records, but also is used as part of methods that infer CO₂ in deep time. The latter case ($\delta^{13}\text{C}$ in models of CO₂ estimation) would create circularity and hold back efforts to produce long-term CO₂ records that are compared to records of temperature. Thus, any manuscript that provides insight into the dependence or independence of $\delta^{13}\text{C}$ of plant material (such as the one here) will be of interest to a wide swath of the deep time paleoanthropological, paleontological, and paleoclimate communities.

The authors utilize a recent high-resolution compilation of carbon dioxide concentrations (and the isotopic value of that atmospheric carbon dioxide) from a suite of ice cores, stretching back 155ka. They use this data to compute predictions of the $\delta^{13}\text{C}$ values of plant organic material using four models of photosynthesis to identify trends in the data for the past 155ka. With two high-resolution datasets for the past ~35ka, they compare how their model fits with the plant and animal data sets, to determine the degree of the effect of atmospheric CO₂ concentration on these two carbon isotope datasets. They use these results to conclude that there is an ~1‰ effect on the isotopic value of plant material per 100 ppm of CO₂ change. This value that is less than that found for speleothem records (1.66‰/100 ppm; Breeker, 2017) and for experimental results on radishes and Arabidopsis plants (up to 2‰/100 ppm; Schubert and Jahren 2012). This is in contrast to a different set

of carbon isotopic records for the same interval from plants and animals (Kohn 2016) that suggest that $\delta^{13}\text{C}$ of plants is effectively independent of CO_2 concentration ($0.0 \pm 0.3 \text{‰}/100 \text{ ppmv}$).

See new Fig. 3, in which we compare (Kohn, 2016) faunal estimates to our own.

The manuscript is well written and the figures are well crafted and very clear. The literature is well cited and up-to-date, and the argument is clearly stated. The authors don't stretch their conclusions beyond the data that they present. Their data is nicely documented in the text and supplemental information.

The novel approach taken here is to determine if there is a dependence of plant $\delta^{13}\text{C}$ on the concentration of CO_2 , by attempting to understand the effect to $p\text{CO}_2$ through models of plant photosynthesis. They use four photosynthesis models (two new, one from Schubert and Jahren, and one modified from Farquhar) to compare their predictions of plant $\delta^{13}\text{C}$ (based upon ice-core data) to standardized values of faunal and plant isotopic data. Commendably, they go to great lengths to standardize their data to the same geographic parameters, which helps their argument. Looking at the data in Figure 3 for fauna I'm fairly convinced that there is a 1‰ shift during the LGM for collagen, and somewhat also convinced for the cellulose record, though there is a large degree of scatter.

However, to my eye, it is the models that seem questionable. None of them are a very good fit to the sample data, so they fall a little short towards supporting the viewpoint that $p\text{CO}_2$ levels are the cause of the $\sim 1\text{‰}$ shift in $\delta^{13}\text{C}_p$. Yes, all models are capable of modeling the anthropogenic influence on $\delta^{13}\text{C}_p$ (except the modified Farquhar 1982 model), but not all of the models are capable of correctly predicting the 1 per mil shift after the LGM. The SJ12 doesn't match the 1 per mil shift (line 124) in plants. The model based upon gymnosperms does match the 1‰ shift, but it fits more poorly for the interval when gymnosperms were the major component of the vegetation (pre-Holocene), and fits better for the part of the record dominated by angiosperms. The angiosperm model has the worst fit to the plant cellulose record, and doesn't even overlap with the part of the record that is dominantly from angiosperm tissue. The $p\text{CO}_2$ independent model (modification of Farquhar) fits the data to a certain degree (though not as well as the SJ12 model) but doesn't match their conclusions of a 1‰ / 100 ppm effect of $p\text{CO}_2$ on $\delta^{13}\text{C}_p$. From their model results, there may be a small effect of $p\text{CO}_2$ levels, but there still remains a significant portion of the data that is not explained by any of the four models.

We agree that none of the models are suitable, but unfortunately a new model would be beyond the scope of our current paper. We included a simulation of each model primarily to help us design our experiment, and to point out that data need to be appropriately chosen to shed light on the $p\text{CO}_2$ effect, since the effects would occur at relatively short periods ($\sim 3\text{-}7\text{kyr}$) in the Quaternary, and dating resolution is therefore crucial. What we now believe is that a good deal of the disagreement is caused by the extent to which both gymnosperm and angiosperm data are included in each particular model. The differences between these groups suggest that both require a different model (as Volker originally claimed), and any future model would have to be specific to fauna/angiosperms or gymnosperms, and either whole wood cellulose/leaf cellulose, where much of the disagreement in absolute $\delta^{13}\text{C}$ arises.

Thus, some other environmental factor, or parameter in the model must be considered before we can use this study to finalize the debate over the effect of $p\text{CO}_2$ on plant carbon isotopic values. Also, there is an insufficient discussion of the changes in source carbon (reservoirs) that occurs during the LGM transition to the Holocene. For instance, I would like to see a comparison of the methane record from the same ice-cores, as this has a much lighter isotopic composition than carbon dioxide, and could explain a considerable amount of the shift towards lighter isotopic composition of the atmospheric CO_2 and therefore the isotopic value of the plants and then the collagen. If the effect of the isotopic reservoir of the carbon is negligible, then it should be

discussed in greater detail and is largely lacking in this manuscript.

We are not exactly sure how methane is directly related to the issue of plant $\delta^{13}\text{C}$ and $p\text{CO}_2$, and it would perhaps be beyond the scope of this manuscript to speculate on this issue. It would indeed be fascinating to compare the sources and sinks of carbon over this period, by comparison with methane isotope records, but we feel that there is already a great deal of information provided, and to include such records might confuse other readers, particularly those who might be approaching the topic from the point of view of palaeoanthropology/archaeology/palaeoecology.

The section on “Prediction of plant isotopes over the last glacial cycle” needs further explanation for us to understand how it has much relevance to the debate over the effect of $p\text{CO}_2$ on $\delta^{13}\text{C}_p$, considering that their comparative sample records don't cover the entire interval of 155ka. They identify three intervals of rapid $\delta^{13}\text{C}_p$ change (12–18, 60–62.7 and 129.4–135 ka), but they only have sample data for the interval of 12–18ka. This section does create a useful set of hypotheses to test, but is only useful for later comparison with high-resolution records (like they have for the LGM) over the 155ka interval. Also, it isn't adequately explained why a change of 0.25‰/ka is considered an important rate. It may be, but as written, this section seems like an output from the modelled ice-core data, rather than supporting evidence for their argument for a dependence of $\delta^{13}\text{C}_{\text{plant}}$ on $p\text{CO}_2$.

Thank you for this suggestion. We have now included a more detailed explanation of the relevance to the debate on $p\text{CO}_2$ – particularly lines 92-102, and rearranged this section to be better incorporated with the section of ice cores. The primary reason for considering rates of change and the timing of these changes is to determine the appropriate dating resolution for testing the $p\text{CO}_2$ effect. We show, in particular, that the Kohn (2016) faunal estimates underestimate the $p\text{CO}_2$ effect due to poor dating resolution, which is an important point of the manuscript.

Reviewers' comments:

Reviewer #1 (Remarks to the Author):

REVISED SUBMISSION

This revised manuscript is improved, most importantly by explicitly considering the effects of MAP and plant type (angiosperm vs. gymnosperm). I found the writing in places to be cluttered and confusing—I lay out some detailed suggestions below. More generally, though, readers will want to know what the average effect of MAP was on the isotopic signal vs. that for atmospheric CO₂. The latter is given in the abstract (and main text), but not the former. This information is needed. It would be interesting to know what the spatial variability is of this partitioning (magnitude of effect by MAP vs. CO₂).

Please supply all data (raw and corrected) in spreadsheet form in the supplement. Maybe I missed a second file, but the raw data are not present in the one supplement file I have.

The authors speak about the relevancy of their analysis for deep-time work (pre-Pleistocene). Given this, they need to speak about the effect of atmospheric O₂ on carbon isotope ratios (it is the O₂/CO₂ ratio that is key). Reference #34 speaks to this issue, and has most of the relevant literature cited within it; interestingly, they find through experiments that O₂ may be more important than CO₂.

Finer-scale comments:

Figure 2B: I understand that the “Cellulose (gymnosperms)” data come only from gymnosperms (squares; as a sidenote—if these all come from *Pinus* leaves, just say that), but what about the “Cellulose” data (circles)? Angiosperms only? Mix of angiosperms and gymnosperms? Wood only?? Also, it's very hard to tell the open circles apart from the open squares, especially in the younger part of the record. Color coding would be helpful. I recommend relegating the “Cellulose” and “Cellulose (adjusted)” data to the supplement, because they aren't particularly helpful for the authors' analyses (and they add a lot of clutter to the figure). Finally, does “adjusted” include latitude, elevation and MAP (similar to panel A)? Please be explicit.

Lines 111-113: One sentence up you say that you are controlling for latitude, altitude and MAP, but based on the next paragraph (starting on line 117), the adjustment you are describing here does not appear to include MAP. Is that correct? More generally, please be clear throughout about what factors you are controlling for. The current language is very confusing.

Line 118: You are also controlling for latitude and elevation, yes?

Lines 122-125: There should be references to Figure 3A in here.

Line 127: “Our fauna primarily reflect a dietary contribution from angiosperms”. Reference please.

Line 138: “(adjusted only for geographical variability)”. I assume this includes latitude, elevation, and MAP??

Line 138: Given that gymnosperms are the only plant group that is common across this transition (line 124), shouldn't this analysis be restricted to just gymnosperms? Ditto for the “plants” analysis presented in line 113. Looking at the open squares in Figure 2B (gymnosperm cellulose), the Voelker gymnosperm model seems to offer the best fit, not SJ-2012.

Lines 138-140: Why are these data not summarized in a supplemental table?

Lines 142-143: "although it may be appropriate for gymnosperms, SJ-2012 should not be used as a baseline to infer changes in angiosperm plants and the majority of ancient fauna". I don't see from the previous sentence why the SJ-2012 is inappropriate for angiosperms.

Line 145: "it best reproduces the magnitude of the pCO₂ effect in our fauna." Which data show this? Table S2 shows that Voelker-2016a has the poorest fit.

Line 155-157: The isotopic offset between leaves and wood cellulose should be folded into the analysis when comparing faunal vs. cellulose patterns. And if you just focus on the Pinus leaf records, this confounding factor disappears.

Lines 210-215: References for equations, please.

Reviewer #3 (Remarks to the Author):

The degree of the effect that pCO₂ has on the isotopic value of fossil plant and faunal material is hotly debated, and this manuscript will tip the balance in favor towards pCO₂ affecting carbon isotopes. It won't end the debate, but this manuscript provides a strong argument that there is an effect, though I believe it needs to be stressed more in the manuscript that this is a maximum value. In particular, environmental factors such as MAP is a key parameters that need better quantification in the future to test the 0.7 ‰ isotopic effect on faunal collagen, and the 1.7 ‰ effect on gymnosperm cellulose. It is clear that the first two reviewers have a better handle on how the authors should include consideration of MAP, so I will leave that discussion up to them.

Reading back through my original comments, and reading the revised manuscript, I now have a better understanding of how the authors were using the physiological models, though I will submit that, as currently written, it will be difficult for the general reader to understand why there isn't a better fit in the relationship between the model output and the cellulose data. Making the procedure used more obvious in the main text (lines 39 to 44), rather than just in the supplementary information, will help readers follow how the ice-core data is used in the models, and then is ultimately compared to the fossil records. The models have the poorest fit to the data in the LGM (and older), and it isn't clear whether it is a fault of the models, or if there is something about the data that, even though corrected, still doesn't match the expected values from the models.

The authors didn't object to my critique of a lack of discussion of the carbon source as a potential mitigating effect, which could potentially be the cause for the isotopic offset present in both datasets at the Pleistocene-Holocene, though they don't add any further discussion of this in the revised text. I thought more about the change in isotopic reservoir, and suggested methane as a potential source. I went and looked, and there is a 300 ppbv increase in CH₄ across this transition (Petit et al., 1999 Science). Methane is potentially -80‰, or ~3x more depleted relative to biological sources of carbon. Since this is just a 0.3 ppm in CH₄ vs. 80 ppm change in CO₂, the effect may be limited, depending upon the isotopic value of the LGM source of CO₂, but it will make the isotopic value of plants more negative.

This ultimately brings me to my general sense for this paper, that the 0.7 ‰ isotopic effect due to pCO₂ is a maximum, and that other factors that are small (CH₄), or hard to quantify (MAP, canopy cover), will add up to the point where there is no isotopic effect, as was found by Kohn (2016) for longer term records. However, I think the authors have answered the criticisms to an extent where I think this should be published, and the debate can continue with this manuscript supporting the argument for an isotopic effect.

Reviewer #1

This revised manuscript is improved, most importantly by explicitly considering the effects of MAP and plant type (angiosperm vs. gymnosperm). I found the writing in places to be cluttered and confusing—I lay out some detailed suggestions below.

More generally, though, readers will want to know what the average effect of MAP was on the isotopic signal vs. that for atmospheric CO₂. The latter is given in the abstract (and main text), but not the former. This information is needed. It would be interesting to know what the spatial variability is of this partitioning (magnitude of effect by MAP vs. CO₂).

Thank you for this suggestion, these are indeed important numbers to provide in the text. We have now included them in the abstract, the main text, and the supplementary (Table S4).

The average effect of MAP on the isotopic signal, constrained by GCMs, is negative for both plants and fauna. The effect on faunal records, which are confined to Eurasia, is -0.40 ± 0.88 ‰, which is larger than both gymnosperms (-0.27 ± 0.55 ‰) and all plants (-0.13 ± 0.74 ‰), but smaller than plants from North America (-0.46 ± 0.88 ‰). It is difficult to compare spatial variability any further, since there are a very limited number of plants from Eurasia at the LGM, and no fauna from North America. However, we can compare fauna (Eurasia) and plants (North America), which we show in the figure below. The corrections are generally higher for North America (yellow boxplots). MAP change in this region is also much more variable, particularly > 20 kyr. In some regions of North America, there is a change towards drier conditions in the Holocene, relative to the LGM. This is also evident from supplementary Figure S5 and Figure 3c. This is the reason for the lower average correction for gymnosperm species of -0.27 ‰, since a significant number of these locations fall within regions which trend towards drier conditions in the Holocene.

Please supply all data (raw and corrected) in spreadsheet form in the supplement. Maybe I missed a second file, but the raw data are not present in the one supplement file I have.

When we submitted the previous corrections, *Nature Communications* did not easily allow for the attachment of a spreadsheet supplementary file, but we have put together a very detailed supplement with all the data and the model outputs in an excel spreadsheet. We now reference this in the "Data Availability" section. Upon publication, the data will be freely available from the *figshare* repository, with an appropriate DOI. Until that time, you are very welcome to access the spreadsheet confidentially via this web link: <https://figshare.com/s/d8c4ee7b8b6f3591118d>

The authors speak about the relevancy of their analysis for deep-time work (pre-Pleistocene). Given this, they need to speak about the effect of atmospheric O₂ on carbon isotope ratios (it is the O₂/CO₂ ratio that is key). Reference #34 speaks to this issue, and has most of the relevant literature cited within it; interestingly, they find through experiments that O₂ may be more important than CO₂.

Thank you for this suggestion - the results of these experiments are indeed interesting, and point to the need for improved comprehensive models of photosynthetic fractionation under atmospheric extrema. We have now included a short discussion of the effects of O₂. Whilst relevant to geological periods of high pCO₂ and/or high pO₂, the importance of the latter is probably minimal in the Quaternary because pO₂ levels changed by a relatively small amount, i.e. ~ 0.7% over the past 800 kyr (Stopler D. A. *et al.*, 2016, *Science*). The short term shifts we observe on millennial timescales during this particular period are therefore more likely due to pCO₂ than pO₂.

Finer-scale comments:

Figure 2B: I understand that the "Cellulose (gymnosperms)" data come only from gymnosperms (squares; as a sidenote—if these all come from *Pinus* leaves, just say that), but what about the "Cellulose" data (circles)? Angiosperms only? Mix of angiosperms and gymnosperms? Wood only?? Also, it's very hard to tell the open circles apart from the open squares, especially in the younger part of the record. Color coding would be helpful. I recommend relegating the "Cellulose" and "Cellulose (adjusted)" data to the supplement, because they aren't particularly helpful for the authors' analyses (and they add a lot of clutter to the figure). Finally, does "adjusted" include latitude, elevation and MAP (similar to panel A)? Please be explicit.

Cellulose (gymnosperms) are not only *Pinus*; there are also species such as Douglas Fir, etc. It is simpler to label them gymnosperms. "Cellulose (adjusted)" referred to either angiosperms or unidentified. We have changed the label to Cellulose (mixed) and Cellulose (mixed), adjusted.

We would prefer to keep the mixed cellulose data in this figure, because it is the majority of our data. It is also how Schubert and Jahren (2012) group their plant taxa. With this in mind, we have improved the legibility of this figure by changing the colour coding; Gymnosperm data are now light and dark blue squares, whereas mixed data are grey and white circles. For clarity, we have changed "adjusted" in both cases to "adjustment for geographic variability", with further information in the figure caption.

Lines 111-113: One sentence up you say that you are controlling for latitude, altitude and MAP, but based on the next paragraph (starting on line 117), the adjustment you are describing here does not appear to include MAP. Is that correct? More generally, please be clear throughout about what factors you are controlling for. The current language is very confusing.

There are two sources of variability which we can reasonably control for: 1) geographic variability in altitude, latitude and MAP, and 2) temporal shifts in MAP across the LGM-Holocene transition. We agree that the terminology is slightly confusing, as it stands. We have changed all instances to either "adjustment for geographic variability" (for 1) OR "corrected for temporal changes in MAP" (for 2) to show the distinction, which is also clarified in the main text.

Line 118: You are also controlling for latitude and elevation, yes?

Yes, that is correct. We have now changed the language to be consistent.

Lines 122-125: There should be references to Figure 3A in here.

We have now included a reference to Figure 3A. Thank you.

Line 127: "Our fauna primarily reflect a dietary contribution from angiosperms". Reference please.

There is an abundant literature of the dietary preferences of specific modern large herbivores, for example; Latham, J. et al., (1999) *Comparative feeding ecology of red (Cervus elaphus) and roe deer (Capreolus capreolus) in Scottish plantation forests*. Journal of Zoology. (<http://onlinelibrary.wiley.com/doi/10.1111/j.1469-7998.1999.tb01003.x/full>)

However, most studies on herbivorous animal ecology are interested in more complex details of dietary contribution. There is no simple single suitable reference to illustrate the range of plant species consumed by the range of our faunal species, particularly in ancient contexts, and so we omit a reference so as not to clutter the text. Although some species are known to be generalists, particularly at times of environmental stress, it is unlikely that any of our fauna had a *significant* dietary contribution from gymnosperm plants. If there was a small contribution, the effect is also likely averaged out in our analysis across both species and time, along with sporadic consumption of isotopically distinct plant tissues: bark, roots, seeds, etc. The only references we could find for species eating significant quantities of anything other than angiosperm plants concern species such as reindeer, which we carefully removed from our analysis.

Line 138: "(adjusted only for geographical variability)". I assume this includes latitude, elevation, and MAP?

Yes, that is correct. We have now clarified this in the text. We hope the explanation given in the previous section makes this clearer.

Line 138: Given that gymnosperms are the only plant group that is common across this transition (line 124), shouldn't this analysis be restricted to just gymnosperms? Ditto for the "plants" analysis presented in line 113. Looking at the open squares in Figure 2B (gymnosperm cellulose), the Voelker gymnosperm model seems to offer the best fit, not SJ-2012.

This is a good suggestion. We included all plant groups in this analysis because the SJ-2012 model is assumed to apply to both angiosperms and gymnosperms, and we originally wished to test the model on the combined dataset, which is what we presumed was appropriate for SJ-2012. We have

now calculated goodness-of-fit statistics for all four models, applied only to the gymnosperm data. We find that SJ-2012 and Voelker-2016g offer similarly good fits (SJ-2012; RMSE=1.07, AIC=25, BIC=31; Voelker-2016g; RMSE=1.04, AIC=26, BIC=34). Ostensibly, Voelker-2016g does indeed yield a marginally better fit in terms of RMSE, but SJ-2012 is better in terms of AIC/BIC, since it is simpler model with greater statistical power. Presently, we would argue that both models are suitable for gymnosperms, but this may well change with the inclusion of well-dated palaeodata in future models. We suspect that many of the plants in the "mixed" category are in fact gymnosperms, since in our experience reliable subfossil $\delta^{13}\text{C}$ from broadleaf species is almost impossible to find at the LGM. When we include these "mixed" data in our analysis, the statistics ultimately favour SJ-2012. However, new data are needed to clarify this. Note that we have also now added these g-o-f statistics to Supplementary Table S2.

Lines 138-140: Why are these data not summarized in a supplemental table?

These data did in fact appear summarised in Supplementary Table S1.

Lines 142-143: "although it may be appropriate for gymnosperms, SJ-2012 should not be used as a baseline to infer changes in angiosperm plants and the majority of ancient fauna". I don't see from the previous sentence why the SJ-2012 is inappropriate for angiosperms.

Thank you for pointing this out. We have now improved the whole paragraph. Further to our clarifications in the main text, it is worthwhile to point out that Schubert and Jahren (2012) included *all* data from Spermatophyte plant groups together in a single regression. The graph below shows the data in Table 1 of that publication, showing the plant type: angiosperm, gymnosperm, or mixed. It is important to note that all data below about 300 ppmv are either gymnosperm (*Pinus flexilis*, *Pinus sylvestris*, *Pinus contortus*, *Sabina przewalskii* and *Juniperus spp.*) or mixed species from palaeo studies. At first glance, angiosperm data appear to conform to the same general relationship as the gymnosperm species, but we would argue that the significant bias towards gymnosperms at low $p\text{CO}_2$ raises the distinct possibility of two separate curves for angiosperms and gymnosperms, as suggested by the Voelker (2016) (perhaps with a steeper hyperbolic relationship for gymnosperm species, which would be consistent with fundamental physiological differences in response to changing c_i/c_a).

Line 145: "it best reproduces the magnitude of the pCO2 effect in our fauna." Which data show this? Table S2 shows that Voelker-2016a has the poorest fit.

Voelker-2016a does have the poorest fit, since the absolute values are offset from the curve, but this model best reproduces the *magnitude* of the deglacial shift, *i.e.* -0.66 ‰ (Table S1), which is closest

to that observed in fauna, -0.53 ‰ (Table S4). A sentence has now been added here for clarification.

Line 155-157: The isotopic offset between leaves and wood cellulose should be folded into the analysis when comparing faunal vs. cellulose patterns. And if you just focus on the Pinus leaf records, this confounding factor disappears.

Multiple authors (e.g. Loader *et al.*, 2003, *Palaeo-3*; Leavitt & Long, 1986, *Ecology*; Rundgren *et al.*, 2003, *J. Quaternary Sci.*) have found that the mean offset between whole wood and cellulose ranges from between 1 ‰ to 2.66 ‰ and slightly higher (3 ‰) between lignin and cellulose. Whole wood most closely represents the total isotopic composition of the leaf sugars. The offsets between whole wood and cellulose/lignin are most likely due to a non-equilibrium fractionation effects during the synthesis of cellulose and lignin from photosynthates.

Considering that our faunal $\delta^{13}\text{C}$ values are most likely a reflection of bulk leaf/whole wood, the range of this offset accounts fairly well for the difference in average collagen-diet enrichment factor over the Holocene. When comparing faunal vs. cellulose patterns, it is important to note that the *Pinus* leaves were still pre-treated to cellulose, and hence should be comparable to other records. We therefore believe that it would not change our estimates of the magnitude of the pCO₂ effect for gymnosperms, when all data are averaged together. Moreover, in terms of comparing faunal vs. cellulose patterns, it is our working assumption that fauna did not eat significant quantities of *Pinus*, as previously mentioned. Upon rereading this section, we believe our statement that "[a] question arises as to why the difference between the cellulose and collagen records, averaged over the Holocene, imply an average collagen-diet enrichment factor ... " might be confusing to the reader, since this is not really a question, but an observation. We have therefore tidied up the main text.

Lines 210-215: References for equations, please.

We have now included a reference to Kohn (2010).

Reviewer #3

The degree of the effect that pCO₂ has on the isotopic value of fossil plant and faunal material is hotly debated, and this manuscript will tip the balance in favor towards pCO₂ affecting carbon isotopes. It won't end the debate, but this manuscript provides a strong argument that there is an effect, though I believe it needs to be stressed more in the manuscript that this is a maximum value. In particular, environmental factors such as MAP is a key parameters that need better quantification in the future to test the 0.7 ‰ isotopic effect on faunal collagen, and the 1.7 ‰ effect on gymnosperm cellulose. It is clear that the first two reviewers have a better handle on how the authors should include consideration of MAP, so I will leave that discussion up to them.

We have inserted a sentence stating that 0.7 ‰ and 1.7 ‰ should be regarded as maxima, if a canopy effect is present. However, there are several good reasons, previously detailed in the main text, which make us think that a canopy effect is unlikely. In the first version of the manuscript our estimates of the pCO₂ effect were indeed upper bounds, because we did not correct for changes in MAP from the LGM to the Holocene. In the current version, MAP changes across the transition are factored into the analysis, which reduces our estimates of the pCO₂ effect. We would also point out that our estimates align reasonably well with recent plant chamber experiments (Voelker-2016), which gives us extra confidence that a canopy effect is unlikely. It is true that other factors such as a canopy effect may have played a role, and from this perspective 0.7 ‰ and 1.7 ‰ should be regarded as maxima, although we have no convincing reason to think that the canopy effect is significant in our records. We suggest that our calculations are the most likely values of the pCO₂ effect, given the best available evidence.

A central point of the paper is that that the effects are less strong than SJ-2012, but stronger than Kohn (2016), who claimed no effect. Once we factor in our detailed analysis of MAP change, we find that the effect is non-zero, using the same regressions as Kohn (2010,2016), but only

considering higher temporal resolution faunal data. The argument we make is that Kohn (2016) did not consider data with sufficient dating resolution to discern shifts due to pCO₂.

Reading back through my original comments, and reading the revised manuscript, I now have a better understanding of how the authors were using the physiological models, though I will submit that, as currently written, it will be difficult for the general reader to understand why there isn't a better fit in the relationship between the model output and the cellulose data. Making the procedure used more obvious in the main text (lines 39 to 44), rather than just in the supplementary information, will help readers follow how the ice-core data is used in the models, and then is ultimately compared to the fossil records. The models have the poorest fit to the data in the LGM (and older), and it isn't clear whether it is a fault of the models, or if there is something about the data that, even though corrected, still doesn't match the expected values from the models.

Thank you for this comment. We have included a great deal more of the procedure in the main text and methods, as well as tried to clarify our language when talking about models, of which there are several: GCMs, plant photosynthetic models. Regarding the fit of the photosynthetic models to the LGM fossil records, we have clarified this section (see comments above to Reviewer #1), and included a short discussion of the probable reasons for this discrepancy. Our perspective is that it is more likely related to the models themselves than the data.

The authors didn't object to my critique of a lack of discussion of the carbon source as a potential mitigating effect, which could potentially be the cause for the isotopic offset present in both datasets at the Pleistocene-Holocene, though they don't add any further discussion of this in the revised text. I thought more about the change in isotopic reservoir, and suggested methane as a potential source. I went and looked, and there is a 300 ppbv increase in CH₄ across this transition (Petit et al., 1999 Science). Methane is potentially -80‰, or ~3x more depleted relative to biological sources of carbon. Since this is just a 0.3 ppm in CH₄ vs. 80 ppm change in CO₂, the effect may be limited, depending upon the isotopic value of the LGM source of CO₂, but it will make the isotopic value of plants more negative.

As far as we know, methane is not a reactant molecule in photosynthesis. Therefore, we have no reason to expect that changes in this part of the reservoir should be directly reflected in the δ¹³C values of plant tissues, since atmospheric CO₂ is taken up by C₃ plants through their stomata and ultimately all carbon in plant tissues is derived from photosynthates which are produced from atmospheric CO₂. Hence, δ¹³C values are dependent on the isotopic composition of the source CO₂, as well as pCO₂ and other environmental effects. It is difficult to see how any other gasses can be involved. With regards to indirect effects, there is no good evidence in the literature for the influence of changing methane levels on plant physiology. In any event, as the reviewer rightly points out, the effect is expected to be insignificant given changes in CH₄ at the ppm level.

This ultimately brings me to my general sense for this paper, that the 0.7 ‰ isotopic effect due to pCO₂ is a maximum, and that other factors that are small (CH₄), or hard to quantify (MAP, canopy cover), will add up to the point where there is no isotopic effect, as was found by Kohn (2016) for longer term records. However, I think the authors have answered the criticisms to an extent where I think this should be published, and the debate can continue with this manuscript supporting the argument for an isotopic effect.

REVIEWERS' COMMENTS:

Reviewer #1 (Remarks to the Author):

The authors have largely addressed my previous concerns.

The new paper in PNAS by Keeling needs to be included (www.pnas.org/cgi/doi/10.1073/pnas.1619240114). They infer a CO₂ effect on plant δ¹³C, but ascribe the mechanism to photorespiration and mesophyll effects; see their equation 1.

Abstract: The magnitude of the CO₂ effect on δ¹³C is given in different units than for the MAP effect. This is confusing because they can't be compared directly. I would report the total effect for CO₂: -0.53 per mil for fauna and around -2.5 per mil for gymnosperms (the latter value is not reported in the main text, so I am making a rough calculation here).

Abstract: The uncertainties on the CO₂ and MAP effects are about as large or larger than the estimated effects. How significant are these effects? At face value, they don't seem significant.

p. 8 top: I don't understand the temporal resolution argument here. Yes, if CO₂ is fluctuating on faster-than-millennial-timescales, I understand the authors' argument. But if CO₂ was more-or-less constant for longer-than-millennial-timescales, for example high CO₂ during the early Paleogene, then the large temporal uncertainties for deep-time records shouldn't matter (much). Also, some CO₂ records come directly from leaves (e.g., stomatal morphology), making—for these records—the argument moot.

p. 10: "The latter discrepancy is due to limitations of that dataset, which is neither large enough nor sufficiently well dated to resolve millennial-scale shifts in δ¹³Cp." This bold statement needs proof or citations.

p. 10: "this photosynthetic model". It's not clear what "this" refers to (as well as "it" in the following sentence).

Reviewer #1

The new paper in PNAS by Keeling needs to be included. They infer a CO₂ effect on plant $\delta^{13}C$, but ascribe the mechanism to photorespiration and mesophyll effects; see their equation 1.

We have added this reference to the concluding section - it nicely illustrates the increasing relevance of models of photosynthetic fractionation for C-cycle and climate change research. As we noted in our Methods Section, and as the Reviewer rightly points out, our version of Farquhar-1982

“is a simplification of an expanded and refined equation [...] which incorporates dissolution of CO₂ in solution, and diffusion inside the leaf, as well as discriminations associated with dark respiration and photorespiration.”

The full equation is used by Keeling et al. (2017) to account for the effect of CO₂ on modern plants. In a very early (pre-submission) version of our manuscript we included these extra terms (Keeling’s Eq. 1), which depend on $1/c_a$, but they did not reproduce the magnitude or direction of the shift in terrestrial archives during the deglaciation. Note that these terms show the opposite sign from fractionations due to carboxylation, and should become more significant at low c_a . The result is intriguing, and further supports our suggestion that the models need revision for use in deep time. Furthermore, the expanded equation, used by Keeling, is arguably too generalised to reproduce what we observe in our data, which is two *different* responses for angiosperms and gymnosperms, a finding also supported by plant data and chamber experiments of Voelker et al. and Porter et al.

Abstract: The magnitude of the CO₂ effect on $\delta^{13}C$ is given in different units than for the MAP effect. This is confusing because they can’t be compared directly. I would report the total effect for CO₂: -0.53 per mil for fauna and around -2.5 per mil for gymnosperms (the latter value is not reported in the main text, so I am making a rough calculation here).

We have decided to give all units in ‰ to make the effect comparable over the deglaciation, as opposed to over a set interval of pCO₂, which is what we originally used [‰ per 100 ppmv] to make our results comparable with Breecker, Kohn and others. These results can be still found in the main text discussion if needed. Note that the value for gymnosperms is actually -1.4 ± 1.2 ‰, and is reported in Supplementary Table 4.

*Abstract: The uncertainties on the CO₂ and MAP effects are about as large or larger than the estimated effects. How significant are these effects? At face value, they don’t seem significant. This is a very interesting question - we thank the reviewer for allowing us to address the subtlety in greater detail. The uncertainties we present in our paper are expanded to include the 1σ range of global plant or faunal stable isotope distribution, as well as uncertainties from ice core records, and uncertainty in MAP change evaluated from ensemble GCMs (detailed in Methods). Therefore, our uncertainties are more conservative than the uncertainties of Kohn (2016), which are given in terms of *s.e.*, and Breecker (2017) who provides 1σ uncertainties (but not expanded). With our uncertainties we intended to show a conservative global range of potential values, and not necessarily the statistical significance of the effect from zero, or a *s.e.*, which is unrealistically small. Note, however, that the*

effect for gymnosperms is, by our measure, still significant at 1σ). Our approach to expanded uncertainties is appropriate because pCO₂ is a global effect which can only be evaluated by partitioning various contributions due to MAP, environment, etc. In other words, C₃ plants exhibit a very large range of scatter due to environmental effects, WUE, etc, and we wished to show this. However, what is of relevance when talking about baseline atmospheric changes in mean d13C over geological timescales is the shift in *average* values, over and above environmental effects. Numerically, the partitioning of the average shift in d13C values is now in agreement with GCMs, and plant chamber experiments which show variable responses from angiosperms and gymnosperms (and a non-zero pCO₂ effect), giving us additional confidence in our result.

p. 8 top: I don't understand the temporal resolution argument here. Yes, if CO₂ is fluctuating on faster-than-millennial-timescales, I understand the authors' argument. But if CO₂ was more-or-less constant for longer-than-millennial-timescales, for example high CO₂ during the early Paleogene, then the large temporal uncertainties for deep-time records shouldn't matter (much). Also, some CO₂ records come directly from leaves (e.g., stomatal morphology), making—for these records—the argument moot.

It is true that during the Paleogene we would not expect large shifts in d13C due to pCO₂, but we maintain that it would not make sense to use these records to discern a pCO₂ effect. We have pointed out at several instances in the manuscript (Figure 1, Supplementary Figure 1, Concluding paragraph, etc.) that most models of photosynthetic fractionation share one thing in common: they predict the greatest change in d13C per ppmv pCO₂ (independent of changes in d13CO₂) at the *lowest* range of pCO₂ (180-300 ppmv). Therefore, during the Palaeogene we would not expect a pCO₂ effect to be visible, hence it makes no sense to use fossil records from these epochs to evaluate a pCO₂ effect. The point of our paragraph is to emphasise the fact that if a pCO₂ effect is present (as predicted by some models), the *only possible* way to resolve the effect is during the last deglaciation, i.e. both when the effect is predicted and when dating methods offer sufficient temporal resolution on sub-millennial timescales. We feel this point is made clear in the manuscript. CO₂ proxies based on stomatal morphology are themselves limited by large uncertainties on predicted pCO₂, typically one to two orders of magnitude (10-100 ppmv) larger than ice core records (e.g. Franks, P.J., et al., *Geophysical Research Letters* 41, 4685–4694, 2014). These uncertainties are arguably too large to discern a pCO₂ effect during the deglaciation. Direct records of ice core pCO₂ and plant d13C are therefore preferred.

p. 10: "The latter discrepancy is due to limitations of that dataset, which is neither large enough nor sufficiently well dated to resolve millennial-scale shifts in 13Cp." This bold statement needs proof or citations.

We are happy to elaborate further. Table S-2 of the Kohn (2016) study presents 98 d13C values from herbivore collagen and tooth enamel, ranging from 11 kyr to 57 Myr. 29 of these data are from the Quaternary, and only 7 of these values, i.e. 7% of the dataset, are from the period between 21 kyr and 11 kyr, when radiocarbon offers sufficient dating to discern the predicted sub-millennial shifts in d13C. We provide several hundred well-dated datapoints over this time range. We also direct the Reviewer to our answer to their previous question.

p. 10: "this photosynthetic model". It's not clear what "this" refers to (as well as "it" in the following sentence).

We have changed these two sentences to refer to the SJ-2012 model explicitly.